# An Empirical Analysis of Uncertainty in Large Language Model Evaluations

**Qiujie Xie**[1,2] **Qingqiu Li**[3] **Zhuohao Yu**[4] **Yuejie Zhang**[3] **Yue Zhang**[2,5] **Linyi Yang**[6,7]†

[1]Zhejiang University [2]School of Engineering, Westlake University [3]School of Computer Science, Shanghai Key Lab of Intelligent Information Processing, Shanghai Collaborative Innovation Center of Intelligent Visual Computing, Fudan University [4]Peking University [5]Westlake Institute for Advanced Study [6]University College London [7]Huawei Noah's Ark Lab

## Abstract

As LLM-as-a-Judge emerges as a new paradigm for assessing large language models (LLMs), concerns have been raised regarding the alignment, bias, and stability of LLM evaluators. While substantial work has focused on alignment and bias, little research has concentrated on the stability of LLM evaluators. In this paper, we conduct extensive experiments involving 9 widely used LLM evaluators across 2 different evaluation settings to investigate the uncertainty in model-based LLM evaluations. We pinpoint that LLM evaluators exhibit varying uncertainty based on model families and sizes. With careful comparative analyses, we find that employing special prompting strategies, whether during inference or post-training, can alleviate evaluation uncertainty to some extent. By utilizing uncertainty to enhance LLM's reliability and detection capability in Out-Of-Distribution (OOD) data, we further fine-tune an uncertainty-aware LLM evaluator named ConfiLM using a human-annotated fine-tuning set and assess ConfiLM's OOD evaluation ability on a manually designed test set sourced from the 2024 Olympics. Experimental results demonstrate that incorporating uncertainty as additional information during the fine-tuning phase can largely improve the model's evaluation performance in OOD scenarios. The code and data are released at: `https://github.com/hasakiXie123/LLM-Evaluator-Uncertainty`.

## 1 Introduction

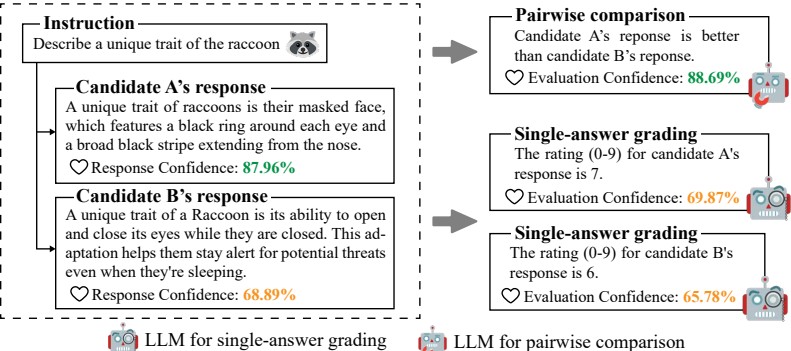

Figure 1: An example of uncertainty (i.e., model confidence) in model-based LLM evaluation. The evaluation process is influenced by the uncertainty of both the evaluator and the candidate model.

Large language models (LLMs) have garnered increasing attention due to their unprecedented performance in various real-world applications (Zhao et al., 2023; Wang et al., 2024a). In this context, how to accurately assess the performance of a LLM becomes particularly important. This area of research includes benchmark-based evaluation, model-based evaluation, and human evaluation (Chang et al., 2024). While various benchmarks (Zellers et al., 2019; Hendrycks et al., 2021; Yang et al., 2023; Xie et al., 2024) have been proposed to measure the core abilities of LLMs in comprehension and generation, human evaluation remains the gold standard for testing overall performance

---

†Correspondence to: Linyi Yang (`yanglinyiucd@gmail.com`).

Figure 2: We conduct extensive experiments and analysis to investigate the existence, mitigation and utilization of uncertainty in model-based LLM evaluation. Uncertainty plays a key role in the evaluation process and can be leveraged to enhance the evaluator's performance in OOD scenarios.

due to its complexity and open-endless. However, this approach is limited by subjectivity issue (Krishna et al., 2023) and resource costs (Karpinska et al., 2021). Consequently, LLM evaluators have emerged as a cost-effective alternative to human evaluators, providing reproducible judgments for responses from different candidate models (Zheng et al., 2023; Wang et al., 2024b; Yu et al., 2024a).

As LLM-as-a-Judge gains more attention, criticism has also emerged (Thakur et al., 2024). Researchers have raised concerns about the alignment (Liu et al., 2024c), bias (Wang et al., 2023a), and stability of model-based LLM evaluation. There has been a surging interest in exploring whether LLM evaluators can truly understand complex contexts and make judgments aligned with human values (Yu et al., 2024a; Hada et al., 2024; Dubois et al., 2024), as well as whether they exhibit preference biases when faced with different inputs (Koo et al., 2023; Liu et al., 2024b; Thakur et al., 2024). Despite significant research on LLM evaluators' alignment and bias, there has been relatively little work on the investigation of evaluation stability. In particular, the relationship between uncertainty and LLM-as-Judge is a question that remains to be underexplored. Can LLMs give consistent evaluation quality across different inputs and domains?

Following previous studies that treat generation logits as a proxy for model confidence (Varshney et al., 2023; Kumar et al., 2024; Duan et al., 2024; Gupta et al., 2024), we use token probabilities to represent the LLM's internal confidence. Through extensive experiments (Figure 2) involving 9 widely-used LLM evaluators under 2 different evaluation settings (single-answer grading and pairwise comparison) on the MT-Bench (Zheng et al., 2023) and PandaLM (Wang et al., 2024b) test sets, we demonstrate that uncertainty is prevalent across LLMs and varies with model families and sizes (§4.2). We find that the evaluation confidence of LLM evaluators exhibits sensitivity to changes in data distribution (§4.3). With careful comparative analyses, we pinpoint that employing special prompting strategies (e.g., chain-of-thoughts; Wei et al. (2022)), whether during inference or post-training, can alleviate evaluation uncertainty to some extent (§4.4 and §4.5).

Prior work has shown that incorporating the model confidence during the LLM's inference stage can improve reliability in OOD scenarios (Yang et al., 2024b) and enhance detection capability in hallucinations (Farquhar et al., 2024). To leverage this fact, we further fine-tune an uncertainty-aware LLM evaluator named ConfiLM using instruction instances collected from the Alpaca 52K dataset (Taori et al., 2023). For evaluation in OOD scenarios (§5), we manually craft a test dataset called Olympic 2024 based on data from the Olympics site. Olympic 2024 contains 220 high-quality instances, each labeled by three PhD-level human evaluators. Samples unanimously deemed low quality by the annotators are removed, resulting in an annotator agreement rate of 97.27%. Experimental results demonstrate that incorporating uncertainty as auxiliary information during the fine-tuning process can largely improve the LLM evaluators' performance in OOD scenarios.

In this paper, we conduct a comprehensive uncertainty analysis, propose a high-quality OOD test set, and offer an uncertainty-aware LLM evaluator named ConfiLM. Our empirical findings reveal the impact of uncertainty on LLM-as-Judge, especially in eliminating and utilizing evaluation uncertainty, shedding light on future research into the stability of model-based LLM evaluations.

## 2 RELATED WORK

With rapid development of LLMs, the accurate evaluation of their capabilities has become one of the key challenges in this field. Several LLM evaluation paradigms have been proposed in recent years (Chang et al., 2024), which have coalesced around a few well-established methods, including benchmark-based evaluation, model-based evaluation, and human evaluation.

**Benchmark-based evaluations** involve using a set of standardized tests to quantitatively measure a model's performance across different tasks. Examples include HellaSwag (Zellers et al., 2019),

HELM (Liang et al., 2022) and MMLU (Hendrycks et al., 2020) for general knowledge and reasoning, or MATH (Hendrycks et al., 2021) and ToolBench (Xu et al., 2023) for specific capabilities. The performance of LLMs is measured by their ability to correctly perform these tasks. However, these metrics often reflect models' performance in narrowly defined areas and risk inflated scores due to data contamination (Oren et al., 2024).

**Human evaluations** involve human raters who assess LLM performance based on criteria such as fluency, coherence, and relevance. This approach can take the form of A/B testing (Tang et al., 2010), preference ranking (Bai et al., 2022), or scoring individual model outputs against predefined rubrics (Novikova et al., 2017). While human evaluations are often considered the gold standard for tasks where quantitative metrics fall short, they are resource-intensive in terms of time and cost. Moreover, they are constrained by subjectivity (Krishna et al., 2023) and reproducible issues (Karpinska et al., 2021), limiting their scalability for large-scale assessments.

**Model-based evaluations** involve employing a powerful LLM as an auto-evaluator to assess the performance of the candidate model. This promising method serves as a cost-effective alternative to human evaluators (Zheng et al., 2023; Wang et al., 2024b; Yu et al., 2024a;b). However, concerns have been raised regarding the alignment, bias, and stability of model-based LLM evaluation. While researchers have made progress in exploring the alignment and bias of LLM evaluators (Liu et al., 2024c; Wang et al., 2023a), understanding the stability of these evaluators remains an open question. A concurrent work (Doddapaneni et al., 2024) proposes a novel framework to evaluate the proficiency of LLM evaluators through targeted perturbations. Different from this work, we focus on the role of uncertainty in LLM-based evaluators, which has yet to be systematically explored.

**Confidence Estimation for LLMs.** Model confidence refers to the degree of certainty a model holds regarding its generated responses (Gal et al., 2016). Reliable confidence estimation for LLM is crucial for effective human-machine collaboration, as it provides valuable insights into the reliability of the model's output, facilitates risk assessment (Geng et al., 2024a), and reduces hallucinations (Varshney et al., 2023). Research in this field includes (1) verbalization-based methods (Lin et al., 2022; Yona et al., 2024), which prompt LLMs to directly output calibrated confidence along with their responses; (2) consistency-based methods (Tian et al., 2023; Xiong et al., 2023), which require LLMs to generate multiple responses for the same question and measure their consistency as a proxy for confidence; and (3) logit-based methods (Duan et al., 2024; Malinin & Gales, 2021; Kumar et al., 2024), which estimate confidence based on the model's internal states during response generation. Inspired by this line of work, we use token probabilities to represent the LLM's internal confidence. Previous work has considered the utilization of model confidence in natural language understanding (Yang et al., 2024b), fact checking (Geng et al., 2024b) and hallucination detection (Varshney et al., 2023; Farquhar et al., 2024). Differently, our work focuses on utilizing confidence within the evaluation process.

## 3 UNCERTAINTY IN LLM-AS-A-JUDGE

**Task definitions.** To ensure the validity of our experimental conclusions, we conduct experiments under two distinct and commonly used evaluation settings, including single-answer grading and pairwise comparison. See Appendix A for the relevant prompts.

(1) Single-answer grading (Yu et al., 2024a; Li et al., 2023; Liu et al., 2023): given a user instruction $q$ and a response $r$ from the candidate model, the evaluator is tasked with assigning an overall score $s \in \mathbb{N}$ based on specific criteria set $c$, while minimizing potential bias. This is expressed as:

$$s = f(q, r; c, \boldsymbol{\theta}), \tag{1}$$

where $c = \{c_1, c_2, \ldots, c_m\}$, each $c_i$ represents a specific evaluation dimension (e.g., content accuracy, logical coherence); $\boldsymbol{\theta}$ represents the parameters of the LLM evaluator.

(2) Pairwise comparison (Wang et al., 2024b; Zeng et al., 2024; Raina et al., 2024): given an instruction $q$ and two responses $r_1$, $r_2$ from different candidate models, the evaluator is asked to compare the two responses and indicate a preference $p \in \{1, 2, \text{Tie}\}$ according to $c$, determining whether one response is better than the other or if they are equally good. This is expressed as:

$$p = f(q, r_1, r_2; c, \boldsymbol{\theta}) \tag{2}$$

**Quantification of uncertainty.** As shown in Figure 1, the LLM-based evaluation process is influenced by the uncertainty of both the evaluator (evaluation uncertainty) and the candidate model (response uncertainty). Following previous studies (Varshney et al., 2023; Zhou et al., 2023b; Gupta et al., 2024), we use token probabilities to represent the LLM's internal confidence. Specifically, we take the probability of the token representing the evaluation result (e.g., "Tie") as the evaluation confidence. For response confidence, we calculate the average probabilities of all generated tokens. See Table 8 for an example of tokens involved in the confidence calculation. To investigate whether different quantification methods impact the empirical findings, we conduct experiments under a pairwise comparison setting on the MT-Bench. The result is presented in Appendix B.1.

# 4 THE IMPACT OF CONFIDENCE IN LLM EVALUATION

We present the empirical study involving 9 widely-used LLM evaluators (3 proprietary models (Achiam et al., 2023) and 6 open-source models (Touvron et al., 2023; Yang et al., 2024a)) with 2 different evaluation settings (single-answer grading and pairwise comparison) on the MT-Bench (Zheng et al., 2023) and PandaLM (Wang et al., 2024b) test datasets.

## 4.1 EXPERIMENTAL SETTINGS

**Prompting strategies.** To explore whether special output formats can reduce the evaluation uncertainty of LLM evaluators, we conduct evaluations using prevalent prompting strategies, including:

(1) **Default** (Wang et al., 2024b; Dubois et al., 2024). We instruct the LLM to act as an impartial judge and consider factors such as helpfulness and relevance. The LLM is asked to first output its rating $s \in \{0, 1, \ldots, 9\}$ or preference $p \in \{1, 2, \text{Tie}\}$, followed by a brief explanation $e$.

(2) Chain-of-thoughts (**CoT**; (Wei et al., 2022; Kojima et al., 2022)). Instead of generating judgments first, we instruct the LLM to first generate a concise reasoning $e$ before providing its rating $s$ or preference $p$ for the responses.

(3) Self-generated reference (**Reference**; (Zheng et al., 2023; Zeng et al., 2024)). We prompt the LLM evaluator to generate a short reference answer $a$ for the given instruction $q$. The generated answer is then provided to the LLM evaluator as a reference when making its judgments.

**LLM Evaluators.** We employ 6 general yet powerful LLMs across various LLM families as evaluators, including GPT-4o (Achiam et al., 2023), GPT-4o-mini, GPT-3.5-Turbo, Llama3-70B-Instruct (Dubey et al., 2024), Llama2-70B-Instruct and Qwen2-72B-Instruct (Yang et al., 2024a). To explore the relationship between evaluation capability and evaluation stability (§4.5), we further assess the stability of 3 specialized LLM evaluators, including (1) **Prometheus2-7b** and **Prometheus2-bgb-8x7b** models (Kim et al., 2024c;b), both of which are trained to output in a **CoT** format, providing a concise rationale before indicating a preference or providing its rating; and (2) **PandaLM** (Wang et al., 2024b), which is trained to output in a default format.

To enhance reproducibility and alleviate the impact of temperature sampling on uncertainty analysis, we set the temperature to 0 for proprietary models, and utilize greedy decoding for open-source models. For single-answer grading, the scoring range is 0-9. The evaluation subject is Llama2-7B-Instruct. For pairwise comparison, the evaluation subjects are Llama2-7B-Instruct and Llama2-13B-Instruct. We query the evaluator twice with the order swapped to eliminate position bias (Wang et al., 2023a; Jung et al., 2019).

**Benchmarks.** We conduct experiments on MT-Bench (Zheng et al., 2023) and PandaLM (Wang et al., 2024b) test dataset. The MT-Bench contains 80 manually constructed questions designed to challenge chatbots based on their core capabilities on common tasks (e.g., reasoning and math). In contrast, the PandaLM test set contains 170 instructions sampled from the human evaluation dataset of self-instruct (Wang et al., 2023b), where expert-written instructions for novel tasks serve as a testbed for evaluating how instruction-based models handle diverse and unfamiliar instructions.

## 4.2 RESULTS AND ANALYSIS

We first conduct an extensive investigation of LLM evaluators with 2 different evaluation settings to gain a preliminary understanding of uncertainty in model-based LLM evaluation, showing partial

Table 1: Uncertainty analysis of 6 LLM-based evaluators on MT-Bench and PandaLM test set. The evaluation subject is Llama2-7B-Instruct and Llama2-13B-Instruct. For single-answer grading, the scoring range is 0-9. "Win / Lose / Tie" represents the average number of times Llama-2-7b-chat's response is better than, worse than, or equal to Llama-2-13b-chat's response.

| Evaluator | Single-answer grading | | | | Pairwise comparison | | | |
| | MT-Bench | | PandaLM Test set | | MT-Bench | | PandaLM Test set | |
| | Rating | Evaluation Confidence | Rating | Evaluation Confidence | Win / Lose / Tie | Evaluation Confidence | Win / Lose / Tie | Evaluation Confidence |
|---|---|---|---|---|---|---|---|---|
| GPT-4o | 5.413 | 0.417 | 6.541 | 0.473 | 10.0 / 16.5 / 53.5 | 0.699 | 30.0 / 38.0 / 102.0 | 0.809 |
| GPT-4o-mini | 6.038 | 0.605 | 6.641 | 0.645 | 27.0 / 43.5 / 9.5 | 0.776 | 53.0 / 61.0 / 56.0 | 0.820 |
| GPT-3.5-Turbo | 6.288 | 0.629 | 6.665 | 0.594 | 38.5 / 35.0 / 6.5 | 0.848 | 76.5 / 81.0 / 12.5 | 0.884 |
| Llama3-70B-Instruct | 7.250 | 0.644 | 7.424 | 0.548 | 39.5 / 37.5 / 3.0 | 0.791 | 78.0 / 86.5 / 5.5 | 0.849 |
| Llama2-70B-Instruct | 7.875 | 0.953 | 7.924 | 0.960 | 33.0 / 34.0 / 13.0 | 0.908 | 72.5 / 73.0 / 24.5 | 0.931 |
| Qwen2-72B-Instruct | 5.875 | 0.675 | 7.153 | 0.692 | 22.0 / 29.0 / 29.0 | 0.762 | 54.0 / 70.0 / 46.0 | 0.806 |
| **Average** | **6.456** | **0.654** | **7.058** | **0.652** | **28.3 / 32.6 / 19.1** | **0.797** | **60.7 / 68.2 / 41.1** | **0.850** |

results in Table 1 for a brief presentation and putting the full results in Appendix D. The following main observations can be drawn:

**LLM evaluators exhibit varying uncertainty based on model families and sizes.** The evaluation uncertainty is more pronounced in the single-answer grading, where the average evaluation confidence is 65.4%, compared to 79.7% for pairwise comparison on MT-Bench. This lower confidence suggests that evaluators exhibit higher uncertainty when scoring individual models, which could stem from evaluators being uncertain about how to score a model's response without the context of a comparison. In contrast, pairwise comparison benefits from direct comparison, leading to more decisive assessments.

**Evaluations within the same model family show significantly higher evaluation confidence.** As shown in Table 1, when Llama2-70B-Instruct is employed to evaluate Llama2-7B-Instruct, both the score (7.875 v.s. 6.456) and evaluation confidence (0.953 v.s. 0.654) are significantly higher than the averages for other evaluators. We speculate that this uncommon high confidence arises from the shared training corpus and similar linguistic patterns between the models, leading to a self-preference bias (Koo et al., 2023; Zheng et al., 2023), where the evaluating model is more familiar with the response style and content generated by a closely related model. This phenomenon highlights the potential threats for self-preference when evaluators from the same model family are used, which could lead to biased evaluations.

**Improved general performance does not guarantee more stable evaluation capabilities.** For example, while GPT-4o demonstrates superior performance in general tasks (such as reasoning and math) compared to GPT-3.5-Turbo (Chiang et al., 2024), its evaluation confidence remains low. In the single-answer grading, GPT-4o has an evaluation confidence of only 0.417, which indicates that despite its enhanced abilities in general tasks, it struggles with stability in evaluating other models' responses. This suggests that there is no certain correlation between a model's competence in performing general tasks and its ability to reliably evaluate the responses of other models, which may be because LLMs are not heavily fine-tuned for the evaluation task (Wang et al., 2024b).

### 4.3 THE INFLUENCES OF DATA DISTRIBUTION

LLMs are typically trained using next token prediction, where the model generates the most likely next word based on the preceding context (Zhao et al., 2023). Different contexts can lead to multiple token choices, and the model makes predictions based on the training distribution, which inherently introduces uncertainty. As displayed in Table 2, we study the impact of data distribution on uncertainty in model-based LLM evaluation. The results demonstrate that evaluation confidence, as measured across both single-answer grading and pairwise comparison settings, exhibits sensitivity to changes in data distribution. When the evaluation scenario shifts from common, high-difficulty tasks (MT-Bench) to novel, unfamiliar tasks (PandaLM test set), the evaluation confidence fluctuates significantly (e.g., from 0.417 to 0.473 on GPT-4o). In contrast, the response confidence (Table 2b) remains more consistent, showing a much smaller variance (**0.014**) between the two datasets. This analysis highlights that in model-based LLM evaluation, evaluation uncertainty is more pronounced compared to response uncertainty, as evidenced by the lower confidence value and larger confidence differences when comparing performance across different datasets.

Table 2: Sensitivity of model confidence to different data distributions. △: the absolute confidence difference between MT-Bench and PandaLM.

(a) Evaluation confidence.

| Model | Single-answer grading | | | Pairwise comparison | | |
|---|---|---|---|---|---|---|
| | MT-Bench | PandaLM Test set | △ | MT-Bench | PandaLM Test set | △ |
| GPT-4o | 0.417 | 0.473 | 0.056 | 0.699 | 0.809 | 0.110 |
| GPT-4o-mini | 0.605 | 0.645 | 0.040 | 0.776 | 0.820 | 0.044 |
| GPT-3.5-Turbo | 0.629 | 0.594 | 0.035 | 0.848 | 0.884 | 0.036 |
| Llama-3-70B-Instruct | 0.644 | 0.548 | 0.096 | 0.791 | 0.849 | 0.058 |
| Llama-2-70B-Instruct | 0.953 | 0.960 | 0.007 | 0.908 | 0.931 | 0.023 |
| Qwen2-72B-Instruct | 0.675 | 0.692 | 0.017 | 0.762 | 0.806 | 0.044 |
| **Average** | **0.654** | **0.652** | **0.042** | **0.797** | **0.850** | **0.053** |

(b) Response confidence.

| Model | MT-Bench | PandaLM Test set | △ |
|---|---|---|---|
| Llama2-7B-Instruct | 0.944 | 0.936 | 0.008 |
| Llama2-13B-Instruct | 0.948 | 0.940 | 0.008 |
| Gemma-1.1-7B-it | 0.867 | 0.858 | 0.009 |
| Qwen2-7B-Instruct | 0.856 | 0.836 | 0.020 |
| Internlm2.5-7B-chat | 0.782 | 0.810 | 0.028 |
| Mistral-7B-Instruct-v0.3 | 0.855 | 0.843 | 0.012 |
| **Average** | **0.875** | **0.871** | **0.014** |

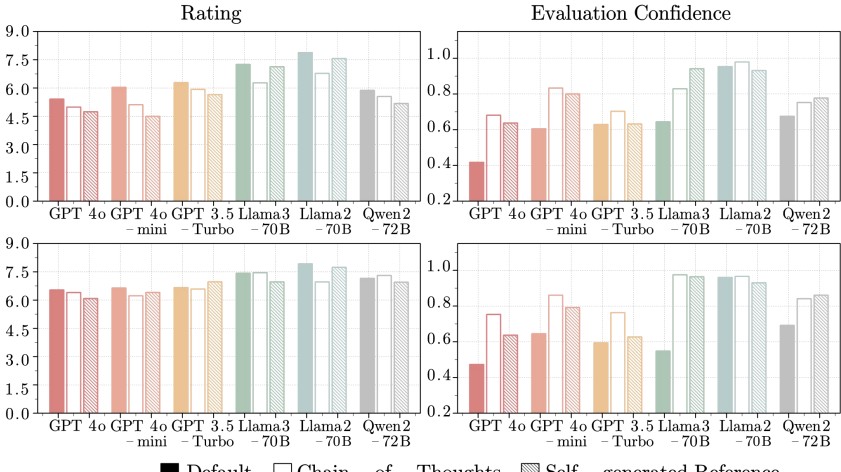

Figure 3: Uncertainty analysis of single-answer grading under special prompting strategies on MT-Bench (first row) and PandaLM Test set (second row). We evaluate Llama2-7B-Instruct with default prompt, chain-of-thoughts and self-generated reference strategies. See Appendix D for full results.

## 4.4 CAN WE EMPLOY PROMPTING STRATEGIES TO MITIGATE UNCERTAINTY?

Prompting is the major approach to solving specialized tasks using LLMs. Prior studies demonstrate that special prompting strategies can enhance LLM's performance on downstream tasks by roleplaying (Salewski et al., 2024), incorporating contextual information (Pan et al., 2023; Yang et al., 2024b) and standardizing output formats (Wei et al., 2022). To explore whether a well-designed prompt can reduce the evaluation uncertainty of LLM evaluators, we conduct experiments using several commonly used prompting strategies, including **Default**, **Chain-of-thoughts** and **Self-generated reference**. The experimental results are shown in Figures 3 and 4. Based on the data presented in Figures 3 and 4, we have the following observations:

(1) Employing special prompting strategies can significantly enhance the evaluation confidence. From the "Evaluation Confidence" subgraphs, we observe that special prompting strategies consistently lead to higher evaluation confidence across different LLM evaluators. In all experiments utilizing the **CoT** strategy, evaluation confidence improved notably. We speculate that this improvement arises from the structured output formats. By explicitly guiding the LLM through step-by-step reasoning before making a judgment, it reduces ambiguity and uncertainty in the evaluation process. While the **Reference** strategy also yields positive results, its effectiveness is less consistent across evaluators, suggesting that the **CoT** strategy is more universally applicable and robust.

(2) The **CoT** strategy seems to alleviate self-preference bias to some extent. For instance, as shown in Figure 3, when Llama2-70B-Instruct evaluates Llama2-7B-Instruct using the **CoT** strategy, the scores are generally lower compared to the **Default** strategy. This decrease indicates that the evaluator, when prompted to generate reasoning first, may become more objective and critical, reducing inherent bias towards the response style and content generated by a closely related model.

(3) Using the **CoT** strategy can enhance the LLM evaluators' abilities to perform fine-grained assessments. As shown in Figure 4, the tie rate decreases in all experiments based on the **CoT** strategy,

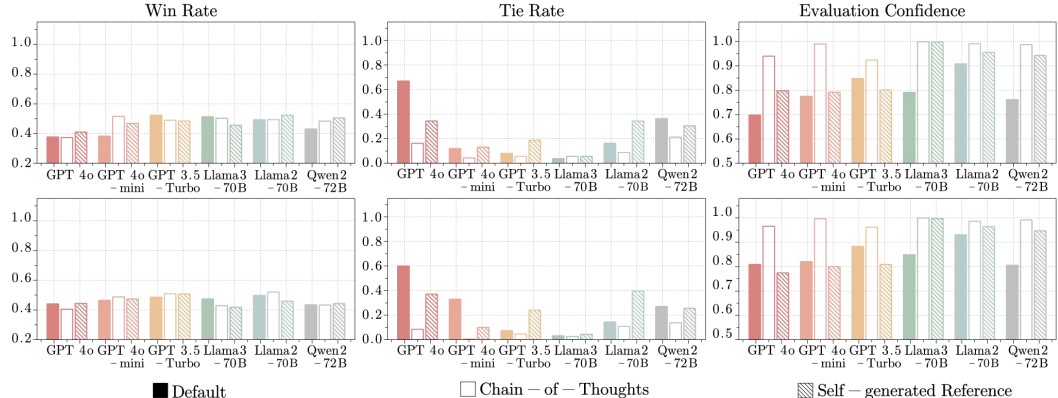

Figure 4: Uncertainty analysis of pairwise comparison under special prompting strategies on MT-Bench (first row) and PandaLM Test set (second row). "Win Rate" represents the proportion of non-tie cases where Llama2-7B-Instruct's response is better than Llama2-13B-Instruct's response. "Tie Rate" represents the proportion of tie cases.

Table 3: Uncertainty analysis with specially trained LLM evaluators on MT-Bench and PandaLM test set. "General LLMs" refers to the average performance of evaluators from Table 1. "Win / Lose / Tie" represents the average number of times Llama2-7B-Instruct's response is better than, worse than, or equal to Llama2-13B-Instruct's response.

(a) Single-answer grading.

| Evaluator | MT-Bench | | PandaLM Test set | |
|---|---|---|---|---|
| | Rating | Evaluation Confidence | Rating | Evaluation Confidence |
| Prometheus2-7B | 5.963 | 0.993 | 7.187 | 0.991 |
| Prometheus2-bgb-8x7B | 4.725 | 0.870 | 6.101 | 0.887 |
| **General LLMs** | **6.456** | **0.654** | **7.058** | **0.652** |

(b) Pairwise comparison.

| Evaluator | MT-Bench | | PandaLM Test set | |
|---|---|---|---|---|
| | Win / Lose / Tie | Evaluation Confidence | Win / Lose / Tie | Evaluation Confidence |
| PandaLM-7B | 42.0 / 28.5 / 9.5 | 0.596 | 58.0 / 72.0 / 40.0 | 0.704 |
| Prometheus2-7B | 37.5 / 42.0 / 0.5 | 0.990 | 77.5 / 92.5 / 0.0 | 0.993 |
| Prometheus2-bgb-8x7B | 31.5 / 32.5 / 16.0 | 0.967 | 77.0 / 80.5 / 12.5 | 0.974 |
| **General LLMs** | **28.3 / 32.6 / 19.1** | **0.797** | **60.7 / 68.2 / 41.1** | **0.850** |

indicating that the evaluator is able to perform fine-grained judgments with the generated rationale, allowing it to distinguish between high-quality responses in complex comparisons. In contrast, although the **Reference** strategy achieves similar effects with GPT-4o and GPT-4o-mini, its benefits are less consistent and not observed across other evaluators.

## 4.5 IS A SPECIALLY TRAINED LLM A MORE STABLE EVALUATOR?

As discussed in Section 4.2, there is still a capability gap between an LLM's general performance and its evaluation ability. Improved general capabilities normally do not guarantee better evaluation capabilities. To address this issue, prior work (Kim et al., 2024a;c; Wang et al., 2024b; Vu et al., 2024) focuses on developing powerful LLM evaluators trained on a large and diverse collection of high-quality human assessments. Are those specially trained LLMs more stable evaluators? We answer this question by experimenting with 3 open-source evaluators including Prometheus2-7b, Prometheus2-bgb-8x7b and PandaLM (Kim et al., 2024c;b; Wang et al., 2024b). The experimental results, as depicted in Table 3, lead to the following conclusions:

(1) The Prometheus2-7b and Prometheus2-bgb-8x7b models, which are trained in a **CoT** format, consistently achieve higher evaluation confidence across all experiments compared to both the General LLMs and the PandaLM. We attribute this phenomenon to the step-by-step rationale provided by the **CoT** strategy, which reduces ambiguity in the evaluation process. This phenomenon aligns with the findings from Section 4.4, confirming that using **CoT** as an output format, whether during inferencing or post-training, can help alleviate evaluation uncertainty in LLM evaluators.

(2) The fine-grained evaluation ability of specially trained LLM evaluators surpasses that of general LLMs, as evidenced by the reduced number of tie cases in pairwise comparison (Table 3b). This improvement is likely due to the incorporation of human assessments as training data, which enhances the evaluators' analytical skills. Moreover, in the Prometheus2 models, this benefit is further amplified by the **CoT** format.

(3) As shown in Table 3a, specially trained LLM evaluators appear to be more sensitive to changes in data distribution. When moving from MT-Bench to the PandaLM test set, the scores of the

Table 4: Data Statistics. The fine-tuning set is sampled from the Alpaca 52K dataset (Taori et al., 2023). Test set (Olympic 2024) is manually created based on data from the Olympics site. Each instance is annotated by three human evaluators.

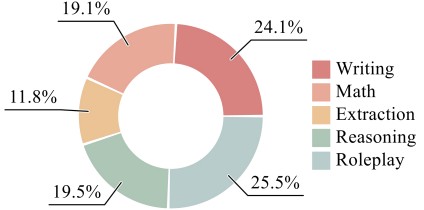

Figure 5: Categories of test instances.

| Data | #Instances | Annotator Agreement |
|---|---|---|
| Fine-tuning set | 694 | 94.96% |
| Test set | 220 | 97.27% |

Prometheus2-7b and Prometheus2-bgb-8x7b models fluctuate more significantly (from 4.725 to 6.101) compared to the general LLMs (from 6.456 to 7.058). Given that Prometheus2-7b and Prometheus2-bgb-8x7b are fine-tuned on specialized data, we speculate that this fluctuation is attributed to the use of teacher forcing in the evaluator's post-training process (Bengio et al., 2015; He et al., 2021), which, while enhancing LLMs' evaluation capabilities, may also increase their sensitivity to changes in data distribution.

Based on the systematic empirical analyses mentioned above, we can conclude that the stability of LLM evaluators is a significant issue, with uncertainty permeating various aspects of model-based LLM evaluation (§4.2). Compared to single-answer grading, pairwise comparison reduces the influence of subjective bias by directly comparing the relative merits of model outputs, thereby mitigating the uncertainty in evaluation to some extent. Furthermore, due to the auto-regressive nature of language models, employing special output formats (such as CoT) can effectively reduce evaluation uncertainty (§4.4 and §4.5). Our findings corroborate the conclusions of Raina et al. (2024) from different perspectives, providing a nuanced analysis of the uncertainty issue.

## 5    MAKING USE OF UNCERTAINTY FOR BETTER EVALUATION

As new and tailored tasks constantly emerge in real applications, they pose OOD challenges (Yang et al., 2023; 2024b; Liu et al., 2024a) to the capability and stability of LLM evaluators. We consider the problem of whether we can utilize the response confidence of candidate models to improve the evaluation capability of LLM evaluators for OOD data. To validate this hypothesis, we first collect ID instances from the Alpaca 52K dataset (Taori et al., 2023) as the fine-tuning set, based on which we fine-tune an uncertainty-aware LLM evaluator named ConfiLM, and assess its evaluation ability on a manually designed OOD test set.

### 5.1    DATASET CONSTRUCTION

**Data collection.** Each instance of the fine-tuning set and OOD test set consists of an input tuple (user instruction $q$, response 1 $r_1$, response confidence of response 1 $u_1$, response 2 $r_2$, response confidence of response 2 $u_2$) and an output tuple (evaluation explanation $e$, evaluation result $p$). Following Wang et al. (2024b), we sample 150 instructions from the Alpaca 52K dataset as the instruction source for the fine-tuning set. For the OOD test set, we manually craft 50 instructions based on data from the Olympics site. We identify 5 common categories of user questions to guide the construction, including writing, math, extraction, reasoning and roleplay. For each category, we then manually design 10 instructions. Each instruction is accompanied by an optional reference answer. We showcase several sample instances and instructions in Tables 9, 10, 11 and 12.

The response pairs $r_1, r_2$ are produced by various instruction-tuned models including Gemma-1.1-7B-it (Team et al., 2024), Internlm2.5-7B-chat (Cai et al., 2024), Qwen2-7B-Instruct (Yang et al., 2024a), and Mistral-7B-Instruct-v0.3 (Jiang et al., 2023). For each source instruction, we pair the responses from two instruction-tuned models, resulting in a total of 900 unprocessed question-response pairs for the fine-tuning set and 300 for the test set. We then employ the calculation method introduced in § 3 to quantify the response confidence $u_1, u_2$. Notably, to ensure the quality and diversity of the generated responses, we set the sampling temperature to 0.7 for all 4 instruction-tuned models. Experimental results (Figure 12) indicate that a sampling temperature of 0.7 achieves comparable response confidence to that of greedy sampling while maintaining generation diversity.

**Human annotations.** The output tuple of each instance includes a brief explanation $e$ for the evaluation and an evaluation result $p$. The evaluation result would be either '1' or '2', indicating that

Table 5: Evaluation performance of 12 evaluators on Olympic 2024. The highest F1 and evaluation confidence of each group is marked by **bold**.

| Evaluator | F1 | | | | | | Evaluation Confidence |
| --- | --- | --- | --- | --- | --- | --- | --- |
| | Overall | Writing | Roleplay | Math | Reasoning | Extraction | |
| GPT-4o | **0.678** | 0.391 | **0.761** | 0.857 | 0.720 | **0.641** | 0.968 |
| GPT-4o-mini | 0.677 | 0.423 | 0.727 | 0.820 | **0.800** | 0.627 | **0.986** |
| GPT-3.5-Turbo | 0.637 | **0.505** | 0.715 | 0.727 | 0.646 | 0.564 | 0.977 |
| Llama3-70B-Instruct | 0.542 | 0.316 | 0.627 | 0.647 | 0.684 | 0.377 | 0.981 |
| Llama2-70B-Instruct | 0.534 | 0.241 | 0.701 | 0.546 | 0.567 | 0.613 | 0.973 |
| Qwen2-72B-Instruct | 0.631 | 0.404 | 0.689 | **0.867** | 0.696 | 0.472 | 0.978 |
| Prometheus2-7B | 0.515 | 0.280 | **0.703** | 0.537 | 0.611 | 0.307 | **0.971** |
| Prometheus2-bgb-8x7B | 0.556 | **0.394** | 0.658 | **0.641** | **0.696** | 0.267 | 0.965 |
| PandaLM-7B | **0.560** | 0.388 | 0.677 | 0.585 | 0.608 | **0.455** | 0.712 |
| Llama3-8B-Instruct | 0.536 | 0.267 | **0.650** | **0.693** | 0.635 | 0.388 | 0.973 |
| Llama3-8B-Instruct-Finetune | 0.582 | 0.603 | 0.573 | 0.333 | 0.628 | 0.458 | 0.979 |
| ConfiLM | **0.621** | **0.723** | 0.566 | 0.510 | **0.670** | **0.594** | **0.982** |

response 1 or response 2 is better. To ensure the quality of human annotations, we involve three experts to concurrently annotate the same data point during the annotation process. These experts are hired by an annotation company, and all annotators receive redundant labor fees. To guarantee clarity and consistency, we provide comprehensive guidelines for every annotator, which emphasizes the need to consider the correctness, logical coherence, vividness and confidence of each response.

**Data preprocessing.** To ensure the quality of the instances and the consistency of human annotations, we implement several data cleaning measures, including (1) removing instances that are unanimously deemed low quality or difficult to evaluate by the annotators; (2) excluding special tokens in the responses (e.g., `<|im_end|>`, `<eos>` ) that may introduce bias to the evaluators; (3) adjusting the ratio of label 1 to label 2 to prevent class imbalance. Additionally, given the ongoing concerns about LLMs' numerical understanding (Liu & Low, 2023), we verbalize each instance's $u_1$ and $u_2$ into natural language statements to avoid introducing additional errors. The ablation result of verbalization is presented in Table 21. The mapping relationship between confidence values and declarative statements is displayed in Table 7. Ultimately, we obtain a fine-tuning set containing 694 high-quality instances and an OOD test set with 220 diverse instances. The annotator agreement rates (Table 4) are 94.96% and 97.27%, respectively. We report the distributions of response confidence $u_1$ and $u_2$ from the finetuning set in Figure 10.

## 5.2 TRAINING DETAILS

Based on the collected fine-tuning set, we fine-tune the Llama3-8B-Instruct by incorporating response confidence as additional information in the prompt (Figure 9), obtaining an uncertainty-aware LLM evaluator named ConfiLM. During the fine-tuning phase of ConfiLM, we use the AdamW (Loshchilov, 2017) optimizer with a learning rate of 5e-5 and a cosine learning rate scheduler. The model is fine-tuned for 6 epochs on 2 NVIDIA A100-SXM4-80GB GPUs. Notably, to differentiate the effects of fine-tuning and the incorporation of response confidence on the model's evaluation performance in the OOD test set, we remove the response confidence $u_1$ and $u_2$ from all fine-tuning instances and fine-tune the Llama3-8B-Instruct again using the same configuration, with the learning rate set to 3e-5. We refer to this variant model as Llama-3-8B-Instruct-Finetune. The performance comparison results between different fine-tuning hyperparameters are presented in Figure 11.

## 5.3 EXPERIMENTAL SETTINGS

To enhance reproducibility, we set the temperature to 0 for proprietary models and utilize greedy decoding for open-source models. For each evaluation, we query the evaluator twice with the order swapped. All general LLM-based evaluators (e.g., GPT-4o) are required to output in a CoT format. To obtain the best evaluation results, specially trained or fine-tuned evaluators (e.g., PandaLM-7B) are assessed using their original prompt and output format.

## 5.4 EVALUATION PERFORMANCE ON OUT-OF-DISTRIBUTION DATA

Table 5 presents the evaluation performance of 12 evaluators on Olympic 2024. Our observations include: (1) all LLM evaluators struggle with the Olympic 2024 (with the best F1 score only reaching

0.678), demonstrating that OOD data poses significant challenges to LLM evaluators' capabilities. (2) ConfiLM outperforms Llama3-8B-Instruct-Finetune and Llama3-8B-Instruct on F1 by 3.9% and 8.5%, respectively. This improvement demonstrates that fine-tuning high-quality human assessments enhances LLMs' evaluation capabilities, and incorporating uncertainty as auxiliary information significantly boosts evaluator performance in OOD scenarios. (3) Compared to reasoning and math tasks, most evaluators show weaker performance on writing tasks. We speculate that this unusual trend arises because LLMs can evaluate response from reasoning tasks based on in-distribution knowledge, but fail to make judgement in creative tasks like writing.

**Case study.** Due to the presence of subtle hallucinations in long texts and the inherently subjective nature of their evaluation, general LLM-based evaluators (such as GPT-4) tend to underperform in writing tasks. We present a test sample (Table 6) that illustrates the role of response confidence in detecting hallucinations of model response (Farquhar et al., 2024; Varshney et al., 2023). As an uncertainty-aware evaluator, ConfiLM reduces the reliability of a response when it detects low confidence, leading to more accurate judgments.

Table 6: Hallucination case. Full version in Table 13.

| |
| --- |
| Generate a news report: Australia defeated Ireland 40:7 in the Women's Rugby Sevens Quarterfinal on 29/07/2024, securing a Semifinal spot. |
| **Response 1:** Women's Rugby Sevens: Australia Cruises Past Ireland in ... (Confidence: 0.865) 
 **Response 2:** ... match between Australia and Ireland took place on 30th July 2024 ... (Confidence: 0.715) |
| **GPT-4o:** Response 2 offers an engaging report. 
 **Llama3-8B-Instruct-Finetune:** Response 2 is detailed. 
 **ConfiLM:** Response 2 contains incorrect dates. |

## 6  DISCUSSION

LLM-based evaluation requires a comprehensive consideration of prompt optimization (Zhou et al., 2023a; 2024a), bias calibration (Zhou et al., 2024b), and uncertainty mitigation strategies. The performance of LLMs as evaluation tools is influenced by various factors, such as the diversity of training data (Shi et al., 2024), inherent model biases (Zheng et al., 2023), and the complexity of the tasks. These uncertainties can cause fluctuations in the consistency of evaluation results. Improving the stability of LLM evaluators can decrease the randomness that may arise during the evaluation process, thus providing more accurate and reproducible results (Chiang & Lee, 2023).

While our work provides extensive analysis on the stability of LLM evaluators, there are other critical aspects of evaluation uncertainty that warrant attention. For example, the relationship between evaluation uncertainty and evaluation bias, as well as the uncertainty in the evaluation of multimodal large language models (Li et al., 2024). Our work only focuses on single-round evaluations. For evaluations conducted on multi-turn benchmarks (i.e., MT-Bench), we use the first-round question as input. It would be interesting to investigate how the uncertainty of LLM evaluators affects judgments on multi-round conversations. Additionally, this research does not cover language models that do not provide token probabilities (e.g., Claude (Anthropic, 2024)). Exploring how to conduct uncertainty analysis for LLM evaluators based on these proprietary models is a valuable topic. It is also important to note that commonly used LLM evaluators require strong calibration to ensure that their output probabilities accurately reflect the precision of their assessments (Chen et al., 2023). We provide an analysis of the relation between evaluation confidence and accuracy in Appendix B.2 and leave further exploration in those aspects to future work.

## 7  CONCLUSION

In this paper, we empirically investigated the existence, mitigation and utilization of uncertainty in model-based LLM evaluation. Extensive empirical analyses demonstrate that uncertainty is prevalent across various LLMs and can be alleviated with special prompting strategies such as chain-of-thought and self-generated reference. Experimental results on an OOD test set with 220 diverse instances show that incorporating uncertainty as auxiliary information during the fine-tuning process can largely improve the LLM evaluators' evaluation performance. We hope the empirical analyses in this work and the proposed uncertainty-aware LLM evaluator can inspire future research on the stability of model-based LLM evaluation.

ACKNOWLEDGMENT

We would like to thank the anonymous reviewers for their insightful comments and suggestions to help improve the paper. This publication has been supported by the National Natural Science Foundation of China (NSFC) Key Project under Grant Number 62336006.

REPRODUCIBILITY STATEMENT

To ensure the reproducibility of our results, we have made detailed efforts throughout the paper. All experimental settings, including model configurations, prompting strategies, and benchmarks, are described in Section §4.1. Additionally, we provide comprehensive information about the dataset construction and training details of ConfiLM in Section §5. Our code, data, and other resources necessary to replicate are released at: `https://github.com/hasakiXie123/LLM-Evaluator-Uncertainty`.

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

## A  PROMPTS DEMONSTRATION

All the relevant prompts used in this study are provided in Figures 6, 7, 8 and 9. Prompts for PandaLM and Prometheus2 model are obtained from their GitHub repository [1].

## B  ANALYSIS

### B.1  DIFFERENT WAYS OF MEASURING UNCERTAINTY

In this paper, we used token probabilities to represent the LLM's internal confidence, a method inspired by previous works (Varshney et al., 2023; Zhou et al., 2023b; Gupta et al., 2024; Kumar et al., 2024). To investigate whether different definitions of uncertainty impact the empirical findings, we conducted additional experiments under a pairwise comparison setting on the MT-Bench dataset. These experiments involved two commonly used confidence quantification methods: (1) Verbalization-based confidence, where we prompted LLMs to directly output calibrated confidence scores along with their responses (Lin et al., 2022; Yona et al., 2024); (2) Consistency-based confidence, which involved generating 5 / 10 / 20 responses to the same question and measuring their consistency as a proxy for confidence (Tian et al., 2023; Xiong et al., 2023). For these experiments, we set the sampling temperature to 0.7.

In the experiments, the evaluation subjects were Llama2-7B-Instruct and Llama2-13B-Instruct. The confidence quantification results are presented in Table 18. Based on the analysis of these results, we observed that the evaluation confidence obtained using different confidence quantification methods follows the same patterns. This further supports the conclusions drawn in Section §4: (1) LLM evaluators exhibit varying levels of uncertainty; (2) Evaluations within the same model family demonstrate higher evaluation confidence.

---

[1]GitHub repository for PandaLM: `https://github.com/WeOpenML/PandaLM`; GitHub repository for Prometheus2 model: `https://github.com/prometheus-eval`.

## B.2 THE RELATION BETWEEN EVALUATION CONFIDENCE AND ACCURACY

To investigate the relation between evaluation confidence and accuracy, we analyzed the average accuracy of judgments made by six LLM-based evaluators on Olympic 2024 across different confidence intervals. The experimental results, as presented in Table 20, reveal a positive correlation between evaluation confidence and accuracy. Specifically, when evaluation confidence is low, the accuracy of judgments across evaluators is generally lower across evaluators. As evaluation confidence increases, judgment accuracy improves steadily, reaching peak performance in high-confidence intervals (e.g., [0.8, 1.0)). This indicates that models are more reliable in performing evaluation tasks when evaluating with higher confidence.

## B.3 THE IN-DOMAIN EVALUATION PERFORMANCE OF CONFILM

In Section §5, we fine-tuned an uncertainty-aware LLM evaluator named ConfiLM, which leverages the response confidence of candidate models to enhance ConfiLM's evaluation capability for OOD data. To investigate the evaluation performance of ConfiLM on in-domain (ID) data, we re-split its fine-tuning dataset, selecting 94 human-annotated instances as an in-domain test set, named Alpaca-94. Based on the remaining 600 fine-tuning instances, we re-trained the models using the same experimental setup as in Section §5.2, obtaining ConfiLM-600 and Llama-3-8B-Instruct-Finetune-600 models. Their evaluation performance on Alpaca-94 (ID data) and Olympic 2024 (OOD data) is reported in Table 19. Experimental results from Table 19 demonstrate that incorporating uncertainty as auxiliary information significantly enhances the performance of LLM evaluators in OOD scenarios. While ConfiLM-600's advantage is reduced in ID scenarios, it still achieves evaluation performance comparable to Llama-3-8B-instruct-finetune-600.

## C DATASET CONSTRUCTION

Each instance of the fine-tuning set and OOD test set consists of an input tuple (user instruction $q$, response 1 $r_1$, response confidence of response 1 $u_1$, response 2 $r_2$, response confidence of response 2 $u_2$) and an output tuple (evaluation explanation $e$, evaluation result $p$). The human-annotated evaluation result would be either '1' or '2', indicating that response 1 or response 2 is better. To ensure the quality and consistency of the human annotations, we first selected 100 samples from the dataset for preliminary annotation by two of the authors. This process facilitated the development of a well-defined annotation guideline. Then, we hired three PhD-level human annotators from an annotation company to annotate all samples (both the fine-tuning set and the test set) in two rounds: (1) In the first round, two annotators were asked to label each sample based on the established annotation guidelines; (2) In the second round, a third annotator reviewed samples where disagreements arose and provided an extra label. The final label for each sample is determined through majority voting. During the annotation process, samples unanimously deemed low quality or difficult to evaluate by the annotators were excluded.

Given the ongoing concerns about LLMs' numerical understanding (Liu & Low, 2023), we verbalized each instance's $u_1$ and $u_2$ into natural language statements to avoid introducing additional errors. The mapping relationship between confidence values and declarative statements is displayed in Table 7. Ultimately, we obtained a fine-tuning set containing 694 high-quality instances and an OOD test set with 220 diverse instances. The annotator agreement rates are 94.96% and 97.27%, respectively. We showcase several sample instances and instructions in Tables 9, 10, 11 and 12.

## D FULL EXPERIMENTAL RESULTS

The full results of experiments introduced in Sections 4 and 5 are displayed in Tables 15, 16 and 17. Additionally, to investigate the impact of response confidence on LLM evaluators' evaluation capabilities, we further conducted experiments under two distinct settings: (1) **default**: providing the evaluator with the complete instance ($q$, $r_1$, $u_1$, $r_2$, $u_2$) and (2) **without confidence**: removing the response confidence $u_1$ and $u_2$ from all test instances. All general LLM-based evaluators (e.g., GPT-4o) were required to output in a CoT format. To obtain the best evaluation results, specially trained or fine-tuned evaluators (e.g., PandaLM-7B) were assessed using their original prompt and output format. Table 14 presents the evaluation performance of 12 evaluators on Olympic 2024.

Based on the analysis of these results, we found that ConfiLM outperforms Llama3-8B-Instruct-Finetune and Llama3-8B-Instruct on F1 by 3.9% and 8.5%, respectively. We attributed this improvement to the incorporation of uncertainty as auxiliary information during the fine-tuning phase. Furthermore, adding uncertainty to the prompts also brings certain performance improvements to general LLM-based evaluators (e.g., 0.690 v.s. 0.641 on GPT-4o-Extraction), but these gains are unstable due to the LLMs' analytical capabilities.

---

**Single-answer grading evaluation**

[Instruction]
Please act as an impartial judge and evaluate the quality of the response provided by an AI assistant to the user question displayed below. Your evaluation should consider factors such as the helpfulness, relevance, accuracy, depth, creativity, and level of detail of the response. You should output an overall score on a scale of 0 to 9, where a higher score indicates better overall performance.
Be as objective as possible. Please avoiding any potential bias. Output your evaluation result and provide a short explanation. The output format is: <The rating is: [Rating]. The explanation is: [Explanation]>.
[Question]
{input}
[The Start of Assistant's Answer]
{output}
[The End of Assistant's Answer]

---

**Single-answer grading evaluation with chain-of-thoughts**

[Instruction]
Please act as an impartial judge and evaluate the quality of the response provided by an AI assistant to the user question displayed below. Your evaluation should consider factors such as the helpfulness, relevance, accuracy, depth, creativity, and level of detail of the response. You should output an overall score on a scale of 0 to 9, where a higher score indicates better overall performance.
Be as objective as possible. Please avoiding any potential bias. You should first provide a short explanation of your evaluation, and then always end your response with your rating. The output format is: <The explanation is: [Explanation]. The rating is: [Rating]>.
[Question]
{input}
[The Start of Assistant's Answer]
{output}
[The End of Assistant's Answer]

---

**Single-answer grading evaluation with self-generated reference**

[Instruction]
Please act as an impartial judge and evaluate the quality of the response provided by an AI assistant to the user question displayed below. Your evaluation should consider factors such as the helpfulness, relevance, accuracy, depth, creativity, and level of detail of the response. You should output an overall score on a scale of 0 to 9, where a higher score indicates better overall performance.
Be as objective as possible. Please avoiding any potential bias. You should first generate a short reference answer for the given question, and then evaluate the AI assistant's answer based on your reference answer, and then always end your response with your rating for the AI assistant's answer. The output format is: <My answer is: [Reference Answer]. Based on the reference answer, my rating for the AI assistant's answer is: [Rating]>.
[Question]
{input}
[The Start of Assistant's Answer]
{output}
[The End of Assistant's Answer]

---

Figure 6: Prompts for single-answer grading. The output format is highlighted in red.

---

**Pairwise comparison**

[Instruction]
Below are two responses for a given task. The task is defined by the user question displayed below with an Input that provides further context. Please act as an impartial judge and evaluate the quality of the responses. You should choose the assistant that follows the user's instructions and answers the user's question better. Your evaluation should consider factors such as the helpfulness, relevance, accuracy, depth, creativity, and level of detail of their responses.
Avoid any position biases and ensure that the order in which the responses were presented does not influence your decision. Do not allow the length of the responses to influence your evaluation. Do not favor certain names of the assistants. Be as objective as possible.
The evaluation result is [1] if first response is better, [2] if second response is better, and [Tie] for a tie. Output your evaluation result and provide a short explanation. The output format is: <The result is: [Evaluation Result]. The explanation is: [Explanation]>.
[Question]
{input}
[The Start of Assistant's Answer]
Response 1: {respponse_1}
Response 2: {respponse_2}
[The End of Assistant's Answer]

---

**Pairwise comparison with chain-of-thoughts**

[Instruction]
Below are two responses for a given task. The task is defined by the user question displayed below with an Input that provides further context. Please act as an impartial judge and evaluate the quality of the responses. You should choose the assistant that follows the user's instructions and answers the user's question better. Your evaluation should consider factors such as the helpfulness, relevance, accuracy, depth, creativity, and level of detail of their responses.
Avoid any position biases and ensure that the order in which the responses were presented does not influence your decision. Do not allow the length of the responses to influence your evaluation. Do not favor certain names of the assistants. Be as objective as possible.
The evaluation result is [1] if first response is better, [2] if second response is better, and [Tie] for a tie. You should first provide a short explanation of your evaluation, and then always end your response with your evaluation result. The output format is: <The explanation is: [Explanation]. The result is: [Evaluation Result]>.
[Question]
{input}
[The Start of Assistant's Answer]
Response 1: {respponse_1}
Response 2: {respponse_2}
[The End of Assistant's Answer]

---

**Pairwise comparison with self-generated reference**

[Instruction]
Below are two responses for a given task. The task is defined by the user question displayed below with an Input that provides further context. Please act as an impartial judge and evaluate the quality of the responses. You should choose the assistant that follows the user's instructions and answers the user's question better. Your evaluation should consider factors such as the helpfulness, relevance, accuracy, creativity, and level of detail of their responses.
Avoid any position biases and ensure that the order in which the responses were presented does not influence your decision. Do not allow the length of the responses to influence your evaluation. Do not favor certain names of the assistants. Be as objective as possible.
The evaluation result is [1] if first response is better, [2] if second response is better, and [Tie] for a tie. You should first generate a short reference answer for the given question, and then evaluate the AI assistants' answers based on your reference answer, and then always end your response with your evaluation result. The output format is: <My answer is: [Reference Answer]. Based on the reference answer, my evaluation result is: [Evaluation Result]>.
[Question]
{input}
[The Start of Assistant's Answer]
Response 1: {respponse_1}
Response 2: {respponse_2}
[The End of Assistant's Answer]

Figure 7: Prompts for pairwise comparison. The output format is highlighted in red.

---

**Single-answer grading evaluation with PandaLM**

[Instruction]
Below are two responses for a given task. The task is defined by the Instruction. Evaluate the responses and generate a reference answer for the task.
### Instruction:
{instruction}
{response}
### Evaluation:

---

**Single-answer grading evaluation with Prometheus2**

###Task Description:
An instruction (might include an Input inside it), a response to evaluate, and a score rubric representing a evaluation criteria are given.
1. Write a detailed feedback that assess the quality of two responses strictly based on the given score rubric, not evaluating in general.
2. After writing a feedback, choose a better response between Response 1 and Response 2. You should refer to the score rubric.
3. The output format should look as follows: "Feedback: (write a feedback for criteria) [RESULT] ([1] or [2] or [Tie])"
4. Please do not generate any other opening, closing, and explanations.
###Instruction: {instruction}
###Response 1: {respponse_1}
###Response 2: {respponse_2}
###Score Rubric: Your evaluation should consider factors such as the helpfulness, relevance, accuracy, depth, creativity, and level of detail of their responses. \n\n Avoid any position biases and ensure that the order in which the responses were presented does not influence your decision. Do not allow the length of the responses to influence your evaluation. Do not favor certain names of the assistants. Be as objective as possible.
###Feedback:

---

**Pairwise comparison with Prometheus2**

###Task Description:
An instruction (might include an Input inside it), a response to evaluate, a reference answer that gets a score of 5, and a score rubric representing a evaluation criteria are given.
1. Write a detailed feedback that assess the quality of the response strictly based on the given score rubric, not evaluating in general.
2. After writing a feedback, write a score that is an integer between 1 and 5. You should refer to the score rubric.
3. The output format should look as follows: "Feedback: (write a feedback for criteria) [RESULT] (an integer number between 1 and 5)
4. Please do not generate any other opening, closing, and explanations.
###The instruction to evaluate {input}
###Response to evaluate: {output}
###Score Rubrics: Your evaluation should consider factors such as the helpfulness, relevance, accuracy, depth, creativity, and level of detail of the response. Be as objective as possible. Please avoiding any potential bias.
###Feedback:

---

Figure 8: Prompts for PandaLM (Wang et al., 2024b) and Prometheus2 model (Kim et al., 2024c). The output format is highlighted in red.

---

**Uncertainty-aware fine-tuning**

[System Prompt]
Below are two responses for a given task. The task is defined by the user question displayed below. Please act as an impartial judge and evaluate the quality of the responses. You should choose the assistant that follows the user's instructions and answers the user's question better. Your evaluation should consider factors such as the helpfulness, relevance, accuracy, depth, creativity, and level of detail of their responses.
Avoid any position biases and ensure that the order in which the responses were presented does not influence your decision. Do not allow the length of the responses to influence your evaluation. Do not favor certain names of the assistants. Be as objective as possible.
The evaluation result is [1] if first response is better, [2] if second response is better. You should first provide a short explanation of your evaluation, and then always end your response with your evaluation result. The output format is: <<The explanation is:\n[Explanation]\nThe result is:[Evaluation Result]>>.
[User Prompt]
User instruction:
<<{instruction}>>
Response 1:
<<{response_1}>>
The confidence of Response 1:
<<{response_1_confidence}>>
Response 2:
<<{response_2}>>
The confidence of Response 2:
<<{response_2_confidence}>>
[Output]
The explanation is:
{evaluation_explanation}
The result is:
{evaluation_result}

Figure 9: Prompts for fine-tuning ConfiLM.

Table 7: The mapping between confidence values and declarative statements.

| Confidence Value | Declarative Statement |
| --- | --- |
| [0, 0.1) | Complete doubt |
| [0.1, 0.2) | Highly uncertain |
| [0.2, 0.3) | Clearly doubtful |
| [0.3, 0.4) | Significantly doubtful |
| [0.4, 0.5) | Slightly doubtful |
| [0.5, 0.6) | Neutral |
| [0.6, 0.7) | Slightly confident |
| [0.7, 0.8) | Clearly confident |
| [0.8, 0.9) | Highly confident |
| [0.9, 1.0] | Absolute confidence |

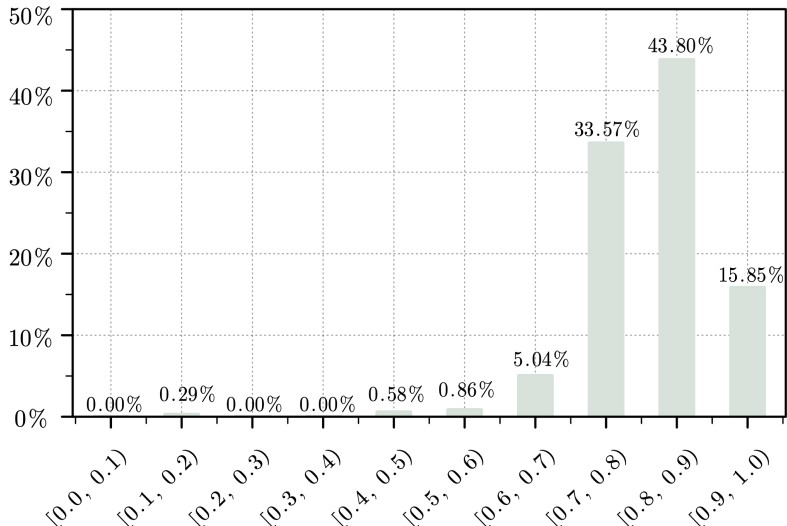

Figure 10: The distribution of response confidence from the fine-tuning set for ConfiLM. The interval [0.0, 0.1) denotes the response confidence is greater than or equal to 0.0 but less than 0.1.

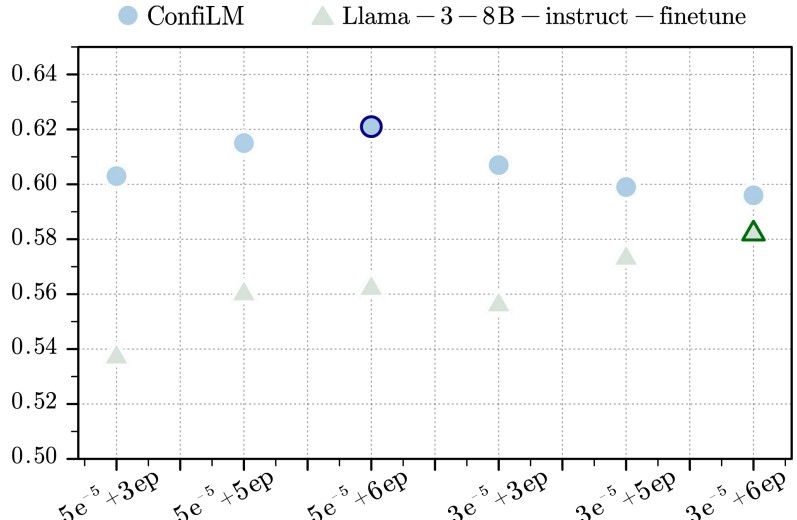

Figure 11: The evaluation performance of ConfiLM and Llama3-8B-Instruct-Finetune under different combinations of learning rate and epoch. $5e^{-5} + 3ep$ represents the combinations of learning rate $5e^{-5}$ and 3 fine-tuning epoch. We selected the hyperparameter combination that yielded the best F1 evaluation performance in Olympic 2024.

Table 8: An example of tokens involved in the confidence calculation. We take the probability of the token representing the evaluation result (e.g., "Tie") as the evaluation confidence. For response confidence, we calculate the average probabilities of all generated tokens. We highlight these tokens with a red background.

| Instruction: Describe a unique trait of the raccoon. | |
|---|---|
| Response 1 | Response 2 |
| A unique trait of raccoons is their masked face, which features a black ring around each eye and a broad black stripe extending from the nose. | A unique trait of a Raccoon is its ability to open and close its eyes while they are closed. This adaptation helps them stay alert for potential threats even when they're sleeping. |

Pairwise comparison for both responses: The evaluation result is 1. The trait provided by Response 2 is incorrect. Response 1 provides the correct answer. Response 1 is better than Response 2.

Single-answer grading for response 1: The rating for response 1 is 7.

Single-answer grading for response 2: The rating for response 2 is 6.

Table 9: A fine-tuning instance for ConfiLM. The fine-tuning set and the test set use the same instance format, which consists of an input tuple (user instruction $q$, response 1 $r_1$, response confidence of response 1 $u_1$, response 2 $r_2$, response confidence of response 2 $u_2$) and an output tuple (evaluation explanation $e$, evaluation result $p$).

| |
|---|
| Instruction $q$: Describe a unique trait of the raccoon. |
| Response 1 $r_1$:
A unique trait of raccoons is their masked face, which features a black ring around each eye and a broad black stripe extending from the nose. |
| Response 1's response confidence $u_1$: Highly confident |
| Response 2 $r_2$:
A unique trait of a Raccoon is its ability to open and close its eyes while they are closed. This adaptation helps them stay alert for potential threats even when they're sleeping. |
| Response 2's response confidence $u_2$: Clearly confident |
| Evaluation explanation $e$:
The trait provided by Response 2 is incorrect. Response 1 provides the correct answer. |
| Preference $p$: 1 |

Table 10: Another fine-tuning instance for ConfiLM.

| |
|---|
| Instruction $q$: What is a good way to handle a difficult problem? |
| Response 1 $r_1$:
How can I provide helpful assistance in resolving it? Please provide me with the specific problem you are facing, and I will do my best to offer suggestions and solutions. |
| Response 1's response confidence $u_1$: Clearly confident |
| Response 2 $r_2$:
Handling a difficult problem can be a challenging task, but here are some steps that might help:1. **Understand the problem**: Clearly define the problem and identify its root cause. Break it down into smaller, manageable parts if necessary. 2. **Research**: Gather ... |
| Response 2's response confidence $u_2$: Highly confident |
| Evaluation explanation $e$:
Response 2 is closer to the user's true intent. Response 1 mistakenly interprets the user's intent as asking for help. |
| Preference $p$: 2 |

Table 11: Examples of user instruction from the Olympic 2024 dataset. Due to space limitations, we truncate the content of the Extraction instance.

**Writing:** Generate a news report based on the following sentences: The Men's Water Polo Gold Medal Match took place on 11/08/2024 at Paris La Defense Arena, Paris. Serbia claimed the Gold with a 13:11 victory against Croatia.

**Math:** In the Women's Synchronised 3m Springboard Final at the Paris Olympics, teams from different countries will compete in five rounds. The scores for each round for the athletes from the United States of America are 49.80, 51.00, 71.10, 72.54, and 70.20. The scores for each round for the athletes from Great Britain are 50.40, 46.20, 63.90, 71.10, and 70.68. What is the total score for the athletes from the United States of America over the five rounds, and on average, how many more points did they score per round compared to the athletes from Great Britain?
Reference answer: 314.64, 2.472.

**Extraction:** I will provide you with a report on a specific Olympic event. Extract the following information from the presented report: the country Sky Brown represents in the Olympics, the sport Sky Brown competes in at the Olympic Games and the age of Sky Brown during the Paris 2024 Olympics. The Report is: For most athletes, the Olympic Games are a battle for medals. For 16-year-old Sky Brown, they are also a battle against injuries. "I don't know what's up with that!" the British skateboarder told Olympics.com about her injury-plagued months leading up to the Olympic Games Paris 2024. "The injury timing is not the best timing. But I do feel like I'm just going to get stronger from this." Brown had to overcome ...
Reference answer: Britain, skateboard, 16.

**Reasoning:** In the men's Group A volleyball competition at the Paris Olympics, each team played against every other team once, and the final results are as follows: Serbia defeated Canada but lost to Slovenia, France lost to Slovenia but won against Canada. Canada lost to Slovenia, and France defeated Serbia. Based on this information, please rank these four teams in their final standings.
Reference answer: Ranking: Slovenia, France, Serbia, Canada.

**Roleplay:** Imagine you are a spectator at an Olympic event, and the athletes have just finished their competition. The team you support has won the competition. A reporter approaches you to ask about your thoughts on the event. I will provide you with a brief report on the event. Based on this information, please respond to the reporter with your impressions of the event and the athlete. The report is: The Men's Water Polo Gold Medal Match took place on 11/08/2024 at Paris La Defense Arena, Paris. Serbia claimed the Gold with a 13:11 victory against Croatia.

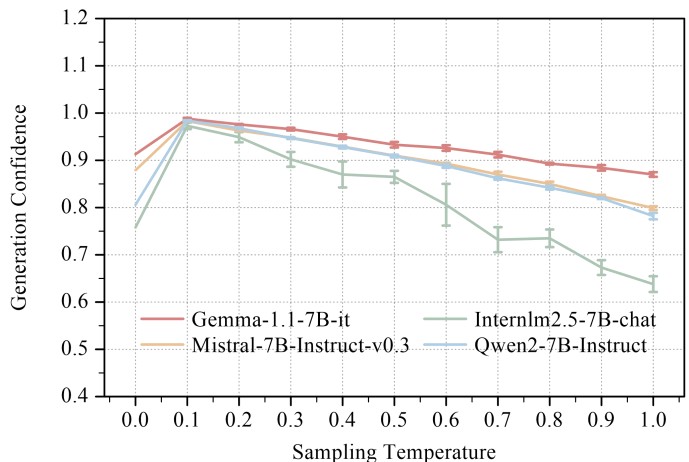

Figure 12: Generation confidence under varying sampling temperatures. We take the average probabilities of all generated tokens as generation confidence and investigate the performance of Gemma-1.1-7B-it (Team et al., 2024), Internlm2.5-7B-chat (Cai et al., 2024), Qwen2-7B-Instruct (Yang et al., 2024a), and Mistral-7B-Instruct-v0.3 (Jiang et al., 2023) on the Olympic2024 test set. To ensure the validity, we run three experiments with the same settings at each sampling temperature.

Table 12: A test instance for ConfiLM.

---

Instruction $q$: In the Women's +81kg weightlifting event at the Paris Olympics, the Chinese athlete successfully lifted 130kg in her first attempt and 136kg in her second attempt in the Snatch category. In the Clean and Jerk category, she successfully lifted 167kg in her first attempt and 173kg in her second attempt, but failed to lift 174kg in her third attempt. The Korean athlete successfully lifted 123kg in her first attempt, 127kg in her second attempt, and 131kg in her third attempt in the Snatch category. In the Clean and Jerk category, she successfully lifted 163kg in her first attempt and 168kg in her second attempt, but failed to lift 173kg in her third attempt. How many more kilograms did the Chinese athlete lift compared to the Korean athlete in their final totals?
Reference answer: 10 kilograms.

---

Response 1 $r_1$:
**Step 1: Calculate the total weight lifted by each athlete.** - Chinese athlete: 136kg (Snatch) + 173kg (Clean and Jerk) = 309kg - Korean athlete: 131kg (Snatch) + 168kg (Clean and Jerk) = 299kg. **Step 2: Calculate the difference in weight lifted between the athletes.** - Difference = 309kg (Chinese) - 299kg (Korean) = **10kg**. Therefore, the Chinese athlete lifted 10kg more than the Korean athlete in their final totals.

---

Response 1's response confidence $u_1$: Absolute confidence

---

Response 2 $r_2$:
To find out how many more kilograms the Chinese athlete lifted compared to the Korean athlete in their final totals, we first need to calculate the total weight lifted by each athlete. **Chinese Athlete:** - Snatch: (130kg + 136kg = 266kg) - Clean and Jerk: (167kg + 173kg = 340kg) - Total: (266kg + 340kg = 606kg) **Korean Athlete:** - Snatch: (123kg + 127kg + 131kg = 381kg) - Clean and Jerk: (163kg + 168kg + 173kg = 504kg) - Total: (381kg + 504kg = 885kg) **Comparison:** The Chinese athlete lifted a total of 606kg, while the Korean athlete lifted a total of 885kg. Therefore, the Korean athlete lifted (885kg - 606kg = 279kg) more than the Chinese athlete.

---

Response 2's response confidence $u_2$: Clearly confident

---

Evaluation explanation $e$:
The answer provided by Response 2 is incorrect. Response 1 provides the correct answer.

---

Preference $p$: 1

---

Table 13: Hallucination case. Response 2 contains a match date that contradicts the instruction.

| |
|---|
| User instruction: Generate a news report based on the following sentences: The Women's Rugby Sevens Quarterfinal between Australia and Ireland was held on 29/07/2024. Australia booked a place in the Semifinals with a 40:7 win. |
| **Response 1:** Women's Rugby Sevens: Australia Cruises Past Ireland in Quarterfinal\n\nThe Australian Women's Rugby Sevens team secured their place in the Semifinals with a dominant 40-7 victory over Ireland in the Quarterfinal on Saturday afternoon. \n\nThe match started with Australia showcasing their attacking prowess, scoring four unanswered tries in the first half. Their clinical execution and strong running game proved too much for Ireland, who struggled to contain the Australian onslaught. \n\nIrish resistance finally came in the second half, with a try of their own. However, it was too little, too late. Australia responded immediately with two more tries to secure their comfortable victory.\n\n\"We are thrilled to advance to the next stage,\" said Australian captain, Bianca McLean. \"The girls played with tremendous spirit and executed their game plan perfectly. We are confident that we can build on this performance and achieve even greater things in the Semifinals.\"\n\nAustralia will face either New Zealand or Fiji in the next round. The winner of that match will go on to contest the final on Sunday.\n\n**Additional Notes:**\n\n* The match took place on 29/07/2024.\n* The score was 40-7 in favor of Australia.\n* Australia booked a place in the Semifinals. |
| **Response confidence 1:** 0.865 |
| **Response 2:** **Title: Australia Triumphs in Women's Rugby Sevens Quarterfinals, Advances to Semifinals**\n\n**Date: 30/07/2024**\n\n**Location: [Host City/Venue]**\n\n**[Host City, July 30, 2024] –** In a thrilling display of rugby prowess and determination, the Women's Rugby Sevens Quarterfinal match between Australia and Ireland took place on 30th July 2024, culminating in a decisive victory for the Australian team.\n\n**Match Overview:**\n\nThe highly anticipated showdown between Australia and Ireland began with a palpable sense of intensity from both sides. The Australian team, renowned for its strength and strategic gameplay, took control early on, setting a robust defensive stance against the Irish team's aggressive tactics.\n\n**First Half Highlight:**\n\nThroughout the first half, Australia's attacking dominance became apparent. [Player Name], a crucial member of the Australian squad, showcased exceptional skills, scoring several crucial tries that paved the way for Australia's substantial lead. The Australian pack also executed precise forward play, creating openings for the backs to exploit.\n\n**Second Half Drama:**\n\nEntering the second half, Ireland displayed remarkable resilience, attempting to claw back the lead through a series of well-executed plays. Their scrum-half [Player Name] managed to breach the Australian defense, scoring a much-needed try for Ireland. However, the Australian team's robust defensive formation and tactical adjustments quickly stifled any further Irish progress.\n\n**Turning Point:**\n\nA critical moment in the game came when [Player Name], an Australian fullback, intercepted an Irish pass, sprinting down the field to score a try. This play, combined with the Australian team's unwavering strategic discipline, sealed the match in favor of Australia.\n\n**Final Score:**\n\nThe match concluded with Australia emerging victorious with a commanding score of 40:7. The resounding |
| **Response confidence 2:** 0.715 |
| **GPT-4o:** The explanation is: Response 2 offers a more thorough and engaging report, despite the missing ending. The result is: [2] |
| **Llama3-8B-Instruct-Finetune:** Explanation: Response 2 is more detailed and considers more aspects. Evaluation result: 2. |
| **ConfiLM:** Explanation: Response 1 is more informative and accurate, and it does not contain factual errors such as incorrect dates. Evaluation result:1. |

Table 14: Evaluation performance on Olympic 2024. The highest F1 and evaluation confidence is marked by **bold**. Green and red are used to indicate whether the evaluator's performance is improved or diminished by the incorporation of model confidence, respectively.

| Evaluator | F1 | | | | | | Evaluation Confidence |
| --- | --- | --- | --- | --- | --- | --- | --- |
| | Overall | Writing | Roleplay | Math | Reasoning | Extraction | |
| GPT-4o | 0.653 | 0.332 | 0.704 | 0.809 | 0.732 | **0.690** | 0.960 |
| w/o confidence | 0.678 | 0.391 | 0.761 | 0.857 | 0.720 | 0.641 | 0.968 |
| GPT-4o-mini | **0.687** | 0.478 | **0.774** | 0.761 | **0.812** | 0.570 | 0.979 |
| w/o confidence | 0.677 | 0.423 | 0.727 | 0.820 | 0.800 | 0.627 | **0.986** |
| GPT-3.5-Turbo | 0.595 | 0.425 | 0.672 | 0.606 | 0.654 | 0.626 | 0.976 |
| w/o confidence | 0.637 | **0.505** | 0.715 | 0.727 | 0.646 | 0.564 | 0.977 |
| Llama3-70B-Instruct | 0.511 | 0.387 | 0.570 | 0.608 | 0.523 | 0.440 | 0.965 |
| w/o confidence | 0.542 | 0.316 | 0.627 | 0.647 | 0.684 | 0.377 | 0.981 |
| Llama2-70B-Instruct | 0.562 | 0.452 | 0.685 | 0.482 | 0.583 | 0.522 | 0.974 |
| w/o confidence | 0.534 | 0.241 | 0.701 | 0.546 | 0.567 | 0.613 | 0.973 |
| Qwen2-72B-Instruct | 0.597 | 0.413 | 0.730 | 0.620 | 0.648 | 0.472 | 0.976 |
| w/o confidence | 0.631 | 0.404 | 0.689 | **0.867** | 0.696 | 0.462 | 0.978 |
| Prometheus2-7B | 0.515 | 0.280 | **0.703** | 0.537 | 0.611 | 0.307 | **0.971** |
| Prometheus2-bgb-8x7B | **0.556** | 0.394 | 0.658 | **0.641** | **0.696** | 0.267 | 0.965 |
| PandaLM-7B | 0.476 | **0.483** | 0.428 | 0.436 | 0.576 | **0.325** | 0.712 |
| Llama3-8B-Instruct | 0.545 | 0.284 | **0.674** | 0.677 | 0.610 | **0.610** | 0.980 |
| w/o confidence | 0.536 | 0.267 | 0.650 | **0.693** | 0.635 | 0.388 | 0.973 |
| Llama3-8B-Instruct-finetune | 0.582 | 0.603 | 0.573 | 0.333 | 0.628 | 0.458 | 0.979 |
| ConfiLM | **0.621** | **0.723** | 0.566 | 0.510 | **0.670** | 0.594 | **0.982** |

Table 15: Full result of uncertainty analysis of single-answer grading on MT-Bench and PandaLM test set. The scoring range is 0-9. The evaluation subject is Llama2-7B-Instruct. The Prometheus2 series models are trained to output in a Chain-of-Thoughts format (providing a concise rationale before indicating a preference between the two outputs) (Kim et al., 2024c).

(a) MT-Bench

| Evaluator | Default | | | Chain-of-Thoughts | | | Self-generated Reference | | |
| --- | --- | --- | --- | --- | --- | --- | --- | --- | --- |
| | Average Rating | Evaluation Confidence | Response Confidence | Average Rating | Evaluation Confidence | Response Confidence | Average Rating | Evaluation Confidence | Response Confidence |
| GPT-4o-2024-05-13 | 5.413 | 0.417 | 0.945 | 4.988 | 0.681 | 0.945 | 4.738 | 0.637 | 0.915 |
| GPT-4o-mini-2024-07-18 | 6.038 | 0.605 | 0.944 | 5.113 | 0.833 | 0.944 | 4.500 | 0.800 | 0.944 |
| GPT-3.5-Turbo | 6.288 | 0.629 | 0.944 | 5.925 | 0.703 | 0.924 | 5.650 | 0.632 | 0.944 |
| Llama-3-70B-Instruct | 7.250 | 0.644 | 0.944 | 6.275 | 0.829 | 0.945 | 7.125 | 0.941 | 0.905 |
| Llama-2-70B-Instruct | 7.875 | 0.953 | 0.945 | 6.775 | 0.979 | 0.915 | 7.563 | 0.931 | 0.925 |
| Qwen2-72B-Instruct | 5.875 | 0.675 | 0.946 | 5.550 | 0.752 | 0.946 | 5.175 | 0.777 | 0.946 |
| Prometheus2-7B | \ | \ | \ | 5.963 | 0.993 | 0.946 | \ | \ | \ |
| Prometheus2-bgb-8x7B | \ | \ | \ | 4.725 | 0.870 | 0.945 | \ | \ | \ |
| Average | 6.456 | 0.654 | 0.945 | 5.486 | 0.886 | 0.942 | 5.792 | 0.786 | 0.930 |

(b) PandaLM Test set

| Evaluator | Default | | | Chain-of-Thoughts | | | Self-generated Reference | | |
| --- | --- | --- | --- | --- | --- | --- | --- | --- | --- |
| | Average Rating | Evaluation Confidence | Response Confidence | Average Rating | Evaluation Confidence | Response Confidence | Average Rating | Evaluation Confidence | Response Confidence |
| GPT-4o-2024-05-13 | 6.541 | 0.473 | 0.935 | 6.400 | 0.753 | 0.935 | 6.082 | 0.637 | 0.934 |
| GPT-4o-mini-2024-07-18 | 6.641 | 0.645 | 0.936 | 6.229 | 0.861 | 0.915 | 6.406 | 0.792 | 0.916 |
| GPT-3.5-Turbo | 6.665 | 0.594 | 0.936 | 6.588 | 0.763 | 0.935 | 6.971 | 0.627 | 0.934 |
| Llama-3-70B-Instruct | 7.424 | 0.548 | 0.937 | 7.453 | 0.975 | 0.907 | 6.965 | 0.964 | 0.927 |
| Llama-2-70B-Instruct | 7.924 | 0.960 | 0.934 | 6.965 | 0.966 | 0.934 | 7.741 | 0.930 | 0.934 |
| Qwen2-72B-Instruct | 7.153 | 0.692 | 0.935 | 7.306 | 0.841 | 0.935 | 6.950 | 0.861 | 0.935 |
| Prometheus2-7B | \ | \ | \ | 7.187 | 0.991 | 0.937 | \ | \ | \ |
| Prometheus2-bgb-8x7B | \ | \ | \ | 6.101 | 0.887 | 0.936 | \ | \ | \ |
| Average | 7.058 | 0.652 | 0.936 | 6.704 | 0.912 | 0.933 | 6.853 | 0.802 | 0.930 |

Table 16: Full result of uncertainty analysis of pairwise comparison on MT-Bench. The evaluation subjects are Llama2-7B-Instruct and Llama2-13B-Instruct. For each evaluation, we query the evaluator twice with the order swapped. 'Win / Lose / Tie' represents the average number of times Llama-2-7b-chat's response is better than, worse than, or equal to Llama-2-13b-chat's response. The PandaLM model (Wang et al., 2024b) is trained to output in a normal format (providing a preference between the two outputs, followed by a concise rationale). The Prometheus2 series models (Kim et al., 2024c) are trained to output in a Chain-of-Thoughts format (providing a concise rationale before indicating a preference between the two responses).

(a) Default prompt

| Evaluator | Default | | | | | |
| | Win | Lose | Tie | Evaluation Confidence | Response Confidence A | B |
|---|---|---|---|---|---|---|
| GPT-4o-2024-05-13 | 10.0 | 16.5 | 53.5 | 0.699 | 0.950 | 0.949 |
| GPT-4o-mini-2024-07-18 | 27.0 | 43.5 | 9.5 | 0.776 | 0.950 | 0.948 |
| GPT-3.5-Turbo | 38.5 | 35.0 | 6.5 | 0.848 | 0.945 | 0.949 |
| Llama-3-70B-Instruct | 39.5 | 37.5 | 3.0 | 0.791 | 0.937 | 0.950 |
| Llama-2-70B-Instruct | 33.0 | 34.0 | 13.0 | 0.908 | 0.950 | 0.950 |
| Qwen2-72B-Instruct | 22.0 | 29.0 | 29.0 | 0.762 | 0.939 | 0.949 |
| PandaLM-7B | 42.0 | 28.5 | 9.5 | 0.617 | 0.949 | 0.939 |
| Average | 35.2 | 30.5 | 14.3 | 0.707 | 0.947 | 0.944 |

(b) Chain-of-Thoughts

| Evaluator | Chain-of-Thoughts | | | | | |
| | Win | Lose | Tie | Evaluation Confidence | Response Confidence A | B |
|---|---|---|---|---|---|---|
| GPT-4o-2024-05-13 | 25.0 | 42.0 | 13.0 | 0.940 | 0.949 | 0.949 |
| GPT-4o-mini-2024-07-18 | 39.5 | 37.0 | 3.5 | 0.990 | 0.950 | 0.948 |
| GPT-3.5-Turbo | 37.0 | 38.5 | 4.5 | 0.924 | 0.950 | 0.949 |
| Llama-3-70B-Instruct | 38.0 | 37.5 | 4.5 | 0.999 | 0.947 | 0.950 |
| Llama-2-70B-Instruct | 36.0 | 37.0 | 7.0 | 0.991 | 0.950 | 0.950 |
| Qwen2-72B-Instruct | 30.5 | 32.5 | 17.0 | 0.988 | 0.949 | 0.949 |
| Prometheus2-7B | 37.5 | 42.0 | 0.5 | 0.990 | 0.949 | 0.951 |
| Prometheus2-bgb-8x7B | 31.5 | 32.5 | 16.0 | 0.967 | 0.948 | 0.950 |
| Average | 34.4 | 37.3 | 8.3 | 0.977 | 0.949 | 0.950 |

(c) Self-generated reference

| Evaluator | Self-generated Reference | | | | | |
| | Win | Lose | Tie | Evaluation Confidence | Response Confidence A | B |
|---|---|---|---|---|---|---|
| GPT-4o-2024-05-13 | 21.5 | 31.0 | 27.5 | 0.799 | 0.949 | 0.949 |
| GPT-4o-mini-2024-07-18 | 32.5 | 37.0 | 10.5 | 0.793 | 0.950 | 0.948 |
| GPT-3.5-Turbo | 31.5 | 33.5 | 15.0 | 0.802 | 0.950 | 0.949 |
| Llama-3-70B-Instruct | 34.5 | 41.0 | 4.5 | 0.991 | 0.947 | 0.950 |
| Llama-2-70B-Instruct | 27.5 | 25.0 | 27.5 | 0.956 | 0.950 | 0.950 |
| Qwen2-72B-Instruct | 28.0 | 27.5 | 24.5 | 0.944 | 0.949 | 0.949 |
| Average | 29.3 | 32.5 | 18.2 | 0.881 | 0.949 | 0.949 |

Table 17: Full result of uncertainty analysis of pairwise comparison and PandaLM test set. The evaluation subjects are Llama2-7B-Instruct and Llama2-13B-Instruct. For each evaluation, we query the evaluator twice with the order swapped. 'Win / Lose / Tie' represents the average number of times Llama-2-7b-chat's response is better than, worse than, or equal to Llama-2-13b-chat's response. The PandaLM model (Wang et al., 2024b) is trained to output in a normal format (providing a preference between the two responses, followed by a concise rationale). The Prometheus2 series models (Kim et al., 2024c) are trained to output in a Chain-of-Thoughts format (providing a concise rationale before indicating a preference between the two responses).

(a) Default prompt

| Evaluator | Default | | | | Response Confidence | |
| --- | --- | --- | --- | --- | --- | --- |
| | Win | Lose | Tie | Evaluation Confidence | A | B |
| GPT-4o-2024-05-13 | 30.0 | 38.0 | 102.0 | 0.809 | 0.932 | 0.941 |
| GPT-4o-mini-2024-07-18 | 53.0 | 61.0 | 56.0 | 0.820 | 0.938 | 0.940 |
| GPT-3.5-Turbo | 76.5 | 81.0 | 12.5 | 0.884 | 0.930 | 0.940 |
| Llama-3-70B-Instruct | 78.0 | 86.5 | 5.5 | 0.849 | 0.941 | 0.941 |
| Llama-2-70B-Instruct | 72.5 | 73.0 | 24.5 | 0.931 | 0.929 | 0.940 |
| Qwen2-72B-Instruct | 54.0 | 70.0 | 46.0 | 0.806 | 0.938 | 0.940 |
| PandaLM-7B | 58.0 | 72.0 | 40.0 | 0.704 | 0.942 | 0.939 |
| Average | 59.3 | 70.1 | 40.5 | 0.777 | 0.938 | 0.940 |

(b) Chain-of-Thoughts

| Evaluator | Chain-of-Thoughts | | | | Response Confidence | |
| --- | --- | --- | --- | --- | --- | --- |
| | Win | Lose | Tie | Evaluation Confidence | A | B |
| GPT-4o-2024-05-13 | 63.0 | 92.5 | 14.5 | 0.966 | 0.940 | 0.940 |
| GPT-4o-mini-2024-07-18 | 82.5 | 86.5 | 1.0 | 0.996 | 0.939 | 0.941 |
| GPT-3.5-Turbo | 82.5 | 79.5 | 8.0 | 0.962 | 0.941 | 0.940 |
| Llama-3-70B-Instruct | 71.0 | 94.5 | 4.5 | 0.999 | 0.941 | 0.941 |
| Llama-2-70B-Instruct | 79.0 | 72.5 | 18.5 | 0.986 | 0.939 | 0.940 |
| Qwen2-72B-Instruct | 63.5 | 83.0 | 23.5 | 0.992 | 0.938 | 0.940 |
| Prometheus2-7B | 77.5 | 92.5 | 0.0 | 0.993 | 0.940 | 0.940 |
| Prometheus2-bgb-8x7B | 77.0 | 80.5 | 12.5 | 0.974 | 0.941 | 0.939 |
| Average | 76.0 | 85.9 | 8.1 | 0.984 | 0.940 | 0.940 |

(c) Self-generated reference

| Evaluator | Self-generated Reference | | | | Response Confidence | |
| --- | --- | --- | --- | --- | --- | --- |
| | Win | Lose | Tie | Evaluation Confidence | A | B |
| GPT-4o-2024-05-13 | 47.5 | 59.5 | 63.0 | 0.774 | 0.940 | 0.940 |
| GPT-4o-mini-2024-07-18 | 72.5 | 80.5 | 17.0 | 0.800 | 0.939 | 0.941 |
| GPT-3.5-Turbo | 65.5 | 63.5 | 41.0 | 0.809 | 0.941 | 0.936 |
| Llama-3-70B-Instruct | 68.0 | 94.5 | 7.5 | 0.997 | 0.941 | 0.941 |
| Llama-2-70B-Instruct | 47.0 | 55.5 | 67.5 | 0.964 | 0.939 | 0.939 |
| Qwen2-72B-Instruct | 56.0 | 70.5 | 43.5 | 0.947 | 0.938 | 0.940 |
| Average | 60.1 | 70.7 | 39.9 | 0.882 | 0.940 | 0.939 |

Table 18: The evaluation confidence results with different quantification methods.

| Evaluator | Logit-based | Verbalization-based | Consistency-5 | Consistency-10 | Consistency-20 |
| --- | --- | --- | --- | --- | --- |
| GPT-4o | 0.699 | 0.764 | 0.751 | 0.725 | 0.696 |
| GPT-4o-mini | 0.776 | 0.725 | 0.771 | 0.736 | 0.735 |
| GPT-3.5-Turbo | 0.848 | 0.755 | 0.828 | 0.790 | 0.804 |
| Llama-3-70B-Instruct | 0.791 | 0.808 | 0.846 | 0.833 | 0.778 |
| Llama-2-70B-Instruct | 0.908 | 0.856 | 0.911 | 0.915 | 0.891 |
| Qwen2-72B-Instruct | 0.762 | 0.730 | 0.765 | 0.717 | 0.657 |
| Average | 0.797 | 0.773 | 0.812 | 0.786 | 0.760 |

Table 19: Evaluation performance on Alpaca-94 and Olympic 2024.

| Dataset | ConfiLM-600 | Llama-3-8B-Instruct-finetune-600 | Llama-3-8B-Instruct |
|---|---|---|---|
| Alpaca-94 | 0.581 | **0.585** | 0.518 |
| Olympic 2024 | **0.577** | 0.535 | 0.519 |

Table 20: The relation between evaluation confidence and evaluation accuracy on Olympic 2024.

| Evaluator | [0.0, 0.2) | [0.2, 0.4) | [0.4, 0.6) | [0.6, 0.8) | [0.8, 1.0) |
|---|---|---|---|---|---|
| GPT-4o | 0.000 | 0.250 | 0.333 | 0.625 | 0.684 |
| GPT-4o-mini | 0.000 | 0.333 | 0.222 | 0.625 | 0.721 |
| GPT-3.5-Turbo | 0.125 | 0.333 | 0.400 | 0.556 | 0.634 |
| Llama-3-70B-Instruct | 0.000 | 0.000 | 0.200 | 0.364 | 0.680 |
| Llama-2-70B-Instruct | 0.000 | 0.000 | 0.000 | 0.267 | 0.579 |
| Qwen2-72B-Instruct | 0.000 | 0.500 | 0.571 | 0.600 | 0.668 |

Table 21: The evaluation performance of ConfiLM with different fine-tuning formats.

| Settings | F1 Score on Olympic 2024 |
|---|---|
| Numerized Confidence | 0.505 |
| Verbalized Confidence | **0.621** |

