# OpenReview forum: "An Empirical Analysis of Uncertainty in Large Language Model Evaluations"
_ICLR.cc/2025/Conference — ICLR 2025 Poster_

### Official Review · Reviewer_2uFv · 2024-10-31

**Soundness:** 3
**Presentation:** 3
**Contribution:** 3
**Rating:** 6
**Confidence:** 5

**Summary:**

The paper mainly discusses uncertainties in evaluations/generations of LLM-based-evaluators. The major contributions of this paper are:
- First, an in-depth evaluation uncertainty of various models across 2 types of tasks (direct-assessment, pairwise-evaluation), 3 prompting strategies (Default, CoT, Self-generated-Reference), 6 LLMs and 2 Benchmarks
- Finetune a new uncertainty aware LLM-evaluator (ConfiLM), that shows lesser uncertainty while evaluating responses.
- The authors also augment the model with a new OOD test set curated from the Olympics website to assess the models for this task.

**Strengths:**

- Firstly, I appreciate the dataset and model created in this paper. These can surely benefit the community in future research in this domain.
- The paper is well-written and provides cogent and coherent arguments till the end.
- The problem discussed in the paper is also quite interesting and important as there is a significant shift towards LLMs as evaluators, however, there are several uncertainties and interpretability issues associated with those judgments, that need to be acknowledged and studied in depth.
- The paper aggregates several tiny established yet undocumented/poorly documented ideas into one place as a formal work.

**Weaknesses:**

- The paper claims to the first to study uncertainties in LLMs as evaluators, which might be incorrect given a previous work already delves in to the same topic of inaccuracies: [Finding Blind Spots in Evaluator LLMs with Interpretable Checklists](https://arxiv.org/abs/2406.13439). I really urge the authors to acknowledge this work in the Related Works section and show the difference with the current work.
- Somehow, I fail to see the use of `Self-Generated Reference`. What is the rationale behind it as compared to `Default` and `CoT`? Also, if the LLM-evaluator has to generate a "good" reference answer, it should first be well-versed with the domain and possess enough knowledge, and this might be difficult of smaller models. For example, the Olympics test set might be OOD for the evaluator as well, and the generated reference answer also might not be optimal, and this might lead to an inaccurate evaluation.
- The paper lacks novelty in some places and feels like a reiteration of existing facts, i.e., most of the findings in section 4.2 are well-known and documented. For example, the second point on self-bias is a long-standing idea and has been proven across multiple tasks and models. Also, there have been several discussions across LLMs excelling at `LMSys ChatBotArena`, however they fail on general tasks. and the last line in the third point of Section 4.2 is also obvious as evaluation is a hard task, and unless models are explicitly trained for it, they will not be suitable for it. Similarly, evaluation is a thorough reasoning task, as the evaluator has to take a proper decision based on the criterion, and input content, so `CoT` performing well in such a case is not a surprise. But, going by my last point in the Strengths, I do not wish to penalize the paper much for this, as these findings are known, but poorly documented, and this paper does a good job of collating them in one single research with appropriate justifications and experiments.
- While building `ConfiLM`, I see that the training instances are < 700 for an 8B model, and FFT was used to train the model. This is slightly concerning as the generalization of the model might be quite less, and there is a good chance of overfitting, especially with 6 epochs. Were any regularizations put into place during this finetuning?

**Questions:**

- When training the second model, `Llama-3-8B-Instruct-Finetune`, why was the learning rate lowered to 3e-5? If anything, for consistency and comparison, the exact same setup as `ConfiLM` should be used, by just removing $u_1$ and $u_2$.
- What happens when $u_1$ and $u_2$ are not verbalized? That would really be an interesting find. Will the model gain more insights from the exact numbers or falter as per previous works? The confidences are converted to discrete classes (Table-8) is slightly dubious as close values on the boundaries of these classes (ex. 0.59, 0.61) see an abrupt jump, which I understand is unavoidable. That's why, a simple experiment without this verbalization should also be studied.
- What is the distribution of $u_1$ and $u_2$ in the finetuning set?
- I feel a very significant portion of the paper is dumped in the appendix. Without referring to the examples in the appendix Table 7-12) the write up about the Olympics Testset and `ConfiLM` evaluation is quite unclear and incomplete. I really urge that content be added to the main paper itself.
- The speculation on line 393 is unclear and needs further support.
- Also, why is the confidence modelled as an arithmetic mean, and not a geometric mean? Geometric mean provides a tighter estimate of the probability, and also goes along with the language-modelling objective.

---

> ### Author Response · Authors · 2024-11-21
> **Response to Reviewer 2uFv**
>
> We appreciate your valuable and positive comments (i.e., **a good job of collating them in one single research with appropriate justifications and experiments.**) In the following, we will carefully respond to your questions.
>
> ### **1. The relation between this work and Finding Blind Spots in Evaluator LLMs with Interpretable Checklists (Weakness 1).**
>
>   - We appreciate the reviewer pointing out the related work "Finding Blind Spots in Evaluator LLMs with Interpretable Checklists". We will incorporate a discussion of this paper into the Related Work section and further clarify the distinctions between our study and theirs.
>   - Specifically, Finding Blind Spots... is an excellent work that proposes FBI, a novel framework designed to evaluate the proficiency of LLM evaluators in assessing four critical abilities (e.g., factual accuracy, instruction following). This work reveals significant shortcomings in current LLM evaluators, but **its primary focus is on testing evaluators' capabilities through targeted perturbations, rather than investigating their behavior when faced with uncertainty or proposing methods for improvement in such scenarios.** Different from their work, our study makes the following key contributions:
>
>     - **Empirical analysis of uncertainty:** we systematically investigate the role of uncertainty in LLM-based evaluators, offering a series of empirical findings that reveal the existence, mitigation and utilization of evaluation uncertainty.
>     - **Human-annotated dataset:** We manually craft a test dataset called Olympic 2024, which contains 220 high-quality instances, each labeled independently by three PhD-level human evaluators. We believe this dataset can serve as a valuable resource for the research community, enabling further exploration into OOD evaluation strategies.
>     - **Uncertainty-aware LLM Evaluator:** We propose an uncertainty-aware evaluator named ConfiLM. The evaluation performance of ConfiLM demonstrates that incorporating uncertainty as auxiliary information can boost evaluator performance in OOD scenarios.
>
> Therefore, unlike Finding Blind Spots in Evaluator LLMs with Interpretable Checklists, our study not only analyzes the uncertainty of LLM Evaluators but also proposes specific methods to leverage and improve the stability of evaluators, particularly their performance in OOD scenarios. We appreciate the reviewer’s suggestion, as it has helped us more comprehensively position our work within the existing literature.

---

> ### Author Response · Authors · 2024-11-21
>
> ### **2. The training details of ConfiLM.**
>
>   - **Selection of Fine-tuning Hyperparameters. (Question 1)**
>     - Thank you for pointing out this issue. We **experimented with multiple hyperparameter combinations and ultimately selected the combination that yielded the best performance in Olympic 2024.** We have listed the evaluation performance of the evaluator fine-tuned with different hyperparameter combinations in Table 1. We include the relevant results in the Appendix to enhance the completeness of the paper.
>     - Table 1: The evaluation performance under different combinations of learning rate and epoch.
>
>         |             Model            | 5e-5 + 3 epoch | 5e-5 + 5 epoch | 5e-5 + 6 epoch | 3e-5+3 epoch | 3e-5+5 epoch | 3e-5+6 epoch |
>         |:----------------------------:|:--------------:|:--------------:|:--------------:|:------------:|:------------:|:------------:|
>         |            ConfiLM           |      0.603     |      0.615     |      **0.621**     |     0.607    |     0.599    |     0.596    |
>         | Llama-3-8B-instruct-finetune |      0.537     |      0.560     |      0.562     |     0.556    |     0.573    |     **0.582**    |
>
>   - **Were any regularizations put into place during the fine-tuning? (Weakness 4)**
>     - To mitigate overfitting during the fine-tuning process, we instructed the annotators to provide as detailed explanations as possible in the evaluation descriptions, which helped enhance the quality of the fine-tuning data. We also used AdamW as the optimizer and experimented with different fine-tuning epochs to identify an optimal setup. Recognizing that using larger data sizes during fine-tuning can improve the model's evaluation performance, **we are actively investigating methods to automatically generate high-quality synthetic data based on the human-annotated dataset provided in this study.** We will release this dataset upon acceptance to help advance future research in this area.
>
>   - **What is the distribution of u1 and u2 in the finetuning set? (Question 3)**
>
>     - We report the distribution of response confidence u1 and u2 from the fine-tuning set in Table 2.
>     - Table 2: The distribution of response confidence from the fine-tuning set.
>         |                | [0.0-0.1) | [0.1-0.2) | [0.2-0.3)  | [0.3-0.4)  | [0.4-0.5) | [0.5-0.6)  | [0.6-0.7) | [0.7-0.8)  | [0.8-0.9)  | [0.9-1.0)  |
>         |:--------------:|:---------:|:---------:|:----------:|:----------:|:---------:|:----------:|:---------:|:----------:|:----------:|:----------:|
>         | Percentage (%) |   0.00    |   0.29    |    0.00    |    0.00    |   0.58    |    0.86    |   5.04    |   33.57    |   43.80    |   15.85    |
>
>   - **Is it necessary to verbalize response confidence u1 and u2? (Question 2)**
>     - This is a great question! In fact, during the fine-tuning of ConfiLM, we tested various strategies for Olympic 2024, including whether to verbalize the response confidence u1 and u2. We present the corresponding experimental results in Table 3. As shown in the results, **verbalization indeed enhanced ConfiLM's evaluation performance.** We believe this improvement is because language models are generally more effective at comparing natural language statements than at working directly with numerical values.
>
>     - Table 3: The evaluation performance of ConfiLM fine-tuned under different formats of response confidence.
>         |  Model  | Numerized Confidence | Verbalized Confidence |
>         |:-------:|:--------------------:|:---------------------:|
>         | ConfiLM |         0.505        |         0.621         |
>
> ### **3. Why did we investigate the uncertainty of LLM evaluator + Self-Generated Reference strategy? (Weakness 2)**
>   - To explore the impact of commonly used prompting strategies on the evaluation uncertainty of LLM evaluators, we conducted experiments using several prompting approaches, including Default, Chain-of-Thoughts, and Self-Generated Reference. We included Self-Generated Reference in our experiments because it is a widely used strategy in LLM evaluation [1][2]. Therefore, we believe it is essential to investigate the uncertainty of evaluator + Self-Generated Reference.
>   - At the same time, we agree with your view that when using smaller LLMs as evaluators, the reference may mislead the evaluator. Hence, in our experimental setup, we chose powerful LLMs as evaluators, such as Llama-3-70B-Instruct, which helps mitigate this issue.

---

> ### Author Response · Authors · 2024-11-21
>
> ### **4. The speculation on line 393 is unclear and needs further support (Question 5).**
>
>   - Thank you for pointing this out. When moving from the MT-Bench to the PandaLM test set, the scores of the Prometheus2-7b and Prometheus2-bgb-8x7b models fluctuate more significantly (from 4.725 to 6.101) compared to the general LLMs (from 6.456 to 7.058). This is an unusual phenomenon, as all evaluation scores are confined to a 0-9 scale, and the fluctuation in scores for the same candidate model should not be this large.
>   - Given that Prometheus2-7b and Prometheus2-bgb-8x7b are evaluators fine-tuned on specialized data, it is possible that the fine-tuning process, particularly the teacher-forcing methods used, may make these models more sensitive to changes in data distribution. We will add a more detailed discussion in this paragraph to clarify this point and improve the readability of the paper.
>
> ### **5. Why use arithmetic mean instead of geometric mean for confidence calculation?  (Question 6)**
>   - Thank you for this constructive suggestion. In this work, we calculated the average probabilities of all generated tokens to represent the response confidence of the candidate model. The reasons for using the arithmetic mean are as follows: (1) The arithmetic mean treats all individual token probabilities equally, ensuring that the aggregation of confidence does not disproportionately emphasize or diminish any token's contribution. (2) The geometric mean, which involves taking the nth root of a product of probabilities, can lead to numerical underflows or excessively low values due to small probabilities, particularly in cases of long outputs.
>
> ### **6. Presentation. (Question 4)**
>   - We will revise the structure of the paper as per your suggestion, with a focus on highlighting Tables 7-12 in the main text.
>
> **Thank you for your professional advice, which helped us make our paper significantly better, strengthening our research work.**
>
> [1] Zheng, L., Chiang, W. L., Sheng, Y., Zhuang, S., Wu, Z., Zhuang, Y., ... & Stoica, I. (2023). Judging llm-as-a-judge with mt-bench and chatbot arena. Advances in Neural Information Processing Systems, 36, 46595-46623.
>
> [2] Zeng, Z., Yu, J., Gao, T., Meng, Y., Goyal, T., & Chen, D. Evaluating Large Language Models at Evaluating Instruction Following. In The Twelfth International Conference on Learning Representations.

---

> > ### Comment · Reviewer_2uFv · 2024-12-02
> > **Final Official Comment to the Authors**
> >
> > Dear Authors,
> >
> > Sorry for the delayed reply, and thank you for your detailed response. Your responses have surely improved my understanding of some parts in the paper. I urge that these points be surely included in the paper, especially the distribution of $u_1$ and $u_2$, and the score difference due to verbalization.
> >
> > However, I feel that I have graded the paper adequately, and I would like to retain my score, but I would surely increase my contribution score to 3. These new points are quite helpful.

---

### Official Review · Reviewer_BN93 · 2024-11-03

**Soundness:** 2
**Presentation:** 3
**Contribution:** 2
**Rating:** 5
**Confidence:** 4

**Summary:**

This paper investigates the uncertainty in LLM-as-Judge evaluations. Specifically, they use the token probabilities for uncertainty estimation, and find that: 1) the uncertainty estimated in this paper is prevalent across LLMs and varies with model families and sizes. 2) the evaluation confidence of LLM evaluators exhibits sensitivity to changes in data distribution. 3) Specific prompting strategies can alleviate evaluation uncertainty to some extent.

**Strengths:**

- Since the paradigm of LLM-as-judge has become widely adopted for evaluation purposes, it is important to conduct an in-depth analysis of the stability of this evaluative framework.
- This paper conducts extensive empirical analyses across multiple models and datasets, offering results that can help researchers deepen their understanding of various models as evaluators.

**Weaknesses:**

- In my opinion, the main issue with this paper is my uncertainty regarding **whether its research methodology adequately supports the conclusions it claims to draw**.
  - As demonstrated in the introduction, the core research question of this paper is "Can large language models provide consistent evaluation quality across different inputs and domains?" However, I am not convinced that the token probabilities predicted by the models sufficiently reflect the "evaluation quality" the authors aim to investigate. For instance, we can see that GPT-4o-mini exhibits significantly higher average prediction probabilities than GPT-4o across various datasets, can we truly infer that GPT-4o-mini possesses superior "evaluation quality"?
  - The token probabilities of large models are influenced by numerous factors, such as the training data, input information, and alignment methods. Relying solely on token probabilities as a measure of uncertainty may affect the reliability and generalizability of the conclusions drawn. A model with a higher output probability is likely just over-confidence.
  - In my opinion, rather than an evaluator that consistently outputs results with high probability, what we require is a well-calibrated evaluator, one whose output probabilities accurately reflect the precision of its assessments.
- The authors claim that this is the first study to investigate uncertainty for LLM-as-judge paradigm. However, many prior studies have also addressed the analysis of evaluation uncertainty, albeit employing different methods for estimating uncertainty. For example, position bias refers to the evaluator providing inconsistent evaluation results when the positions of candidate responses are swapped. This, too, is a method for estimating the uncertainty of an evaluator.

**Questions:**

see weaknesses

---

> ### Author Response · Authors · 2024-11-21
> **Response to Reviewer BN93**
>
> Thank you for your valuable feedback. We carefully address your concerns in specific comments below.
>
> ### **1. The validity of our conclusions (Weakness 1).**
>
>   - In this study, we used token probabilities to represent the LLM's internal confidence, a method inspired by previous works [1, 2]. We understand your concerns about the validity of our conclusions. **To address these concerns, we conducted additional experiments to further substantiate the validity of our findings from different perspectives:** (1) Does using different definitions of uncertainty impact the final findings? (2) What is the relationship between logit-based evaluation uncertainty and evaluation quality?
>
>   - **(1) Does using different definitions of uncertainty impact the final findings?**
>       -  To address this question, we conducted additional experiments under a pairwise comparison setting on the MT-Bench. These experiments involved two commonly used confidence estimation methods: (1) **Verbalization-based confidence**, where we prompted LLMs to directly output calibrated confidence scores along with their responses [3][4];  (2) **Consistency-based confidence**, which involved generating 5/10/20 responses to the same question and measuring their consistency as a proxy for confidence [5][6]. For these experiments, we set the sampling temperature to 0.7.
>       -  In the experiments, the evaluation subjects were Llama2-7B-Instruct and Llama2-13B-Instruct. The confidence estimation results are presented in Table 1. **Based on the analysis of these results, we observed that the evaluation confidence obtained using different confidence estimation methods follows the same patterns.** This further supports the conclusions drawn in the original submission: (1) LLM evaluators exhibit varying levels of uncertainty. (2) Evaluations within the same model family demonstrate higher evaluation confidence.
>
>       -  Table 1: The evaluation confidence estimation results with different estimation methods.
>             |       Evaluator      | Logit-based  | Verbalization-based  | Consistency-5 | Consistency-10 | Consistency-20 |
>             |:--------------------:|:------------:|:--------------------:|:-------------:|:--------------:|:--------------:|
>             |        GPT-4o        |     0.699    |         0.764        |     0.751     |      0.725     |      0.696     |
>             |      GPT-4o-mini     |     0.776    |         0.725        |     0.771     |      0.736     |      0.735     |
>             |     GPT-3.5-Turbo    |     0.848    |         0.755        |     0.828     |      0.790     |      0.804     |
>             | Llama-3-70B-Instruct |     0.791    |         0.808        |     0.846     |      0.833     |      0.778     |
>             | Llama-2-70B-Instruct |     0.908    |         0.856        |     0.911     |      0.915     |      0.891     |
>             |  Qwen2-72B-Instruct  |     0.762    |         0.730        |     0.765     |      0.717     |      0.657     |

---

> ### Author Response · Authors · 2024-11-21
>
> - **(2) What is the relation between logit-based evaluation uncertainty and evaluation quality?**
>
>   - We agree that ''A model with a higher output probability is likely just over-confident''. To further explore the relationship between probability and ability,  we analyzed the average accuracy of judgments made by six LLM-based evaluators across different confidence intervals. The experiments were conducted on our human-annotated test set (Olympic 2024). The results are presented in Table 2. This table demonstrates **a positive correlation between evaluation confidence and evaluation accuracy.** Specifically, when evaluation confidence is low, the accuracy of judgments across evaluators is generally lower. As evaluation confidence increases, judgment accuracy improves steadily, reaching peak performance in high-confidence intervals (e.g., [0.8, 1.0)). This indicates that models are more reliable in performing evaluation tasks when outputting with higher confidence.
>
>   - Table 2: The relation between logit-based evaluation confidence and evaluation accuracy on Olympic 2024.
>
>       |       Evaluator      | [0.0, 0.2) | [0.2, 0.4) | [0.4, 0.6) | [0.6, 0.8) | [0.8, 1.0) |
>       |:--------------------:|:----------:|:----------:|:----------:|:----------:|:----------:|
>       |        GPT-4o        |   0.000    |   0.250    |   0.333    |    0.625   |   0.684    |
>       |      GPT-4o-mini     |   0.000    |   0.333    |   0.222    |    0.625   |   0.721    |
>       |     GPT-3.5-Turbo    |   0.125    |   0.333    |   0.400    |    0.556   |   0.634    |
>       | Llama-3-70B-Instruct |   0.000    |   0.000    |   0.200    |    0.364   |   0.680    |
>       | Llama-2-70B-Instruct |   0.000    |   0.000    |   0.000    |    0.267   |   0.579    |
>       |  Qwen2-72B-Instruct  |   0.000    |   0.500    |   0.571    |    0.600   |   0.668    |
>
> Through the two experiments mentioned above, we demonstrated that **evaluation confidence obtained using different confidence estimation methods exhibits patterns consistent with the conclusions of this study**. Additionally, we observed **a positive correlation between logit-based evaluation confidence and evaluation accuracy**. We will incorporate the results of these experiments into Section 3 and Section C, and submit a new version soon.
>
> ### **2. Contributions of this study (Weakness 2).**
>
>   - We agree with the view that many prior studies have established a strong foundation for understanding the limitations of LLM evaluators, including position bias [7], self-preference bias [8], and misalignment [9]. Building on this foundation, our study makes the following key contributions:
>     - **Empirical analysis of uncertainty:** we systematically investigate the role of uncertainty in LLM-based evaluators, offering a series of empirical findings that reveal the existence, mitigation and utilization of evaluation uncertainty.
>     - **Human-annotated dataset:** We manually craft a test dataset called Olympic 2024, which contains 220 high-quality instances, each labeled independently by three PhD-level human evaluators. We believe this dataset can serve as a valuable resource for the research community, enabling further exploration into OOD evaluation strategies.
>     - **Uncertainty-aware LLM Evaluator:** We propose an uncertainty-aware evaluator named ConfiLM. The evaluation performance of ConfiLM demonstrates that incorporating uncertainty as auxiliary information can boost evaluator performance in OOD scenarios.
>
>   - **We agree with your idea that studying a well-calibrated evaluator is crucial.** As shown in Table 2, commonly used LLM evaluators require stronger calibration to ensure that their output probabilities accurately reflect the precision of their assessments. This will be an important direction for future work in this study and we will add a discussion about this direction in Section 6.
>
> **We look forward to further communication with you and are open to any discussion that can address your doubts.**

---

> ### Author Response · Authors · 2024-11-21
>
> [1] Yang, L., Zhang, S., Yu, Z., Bao, G., Wang, Y., Wang, J., ... & Zhang, Y. Supervised Knowledge Makes Large Language Models Better In-context Learners. In The Twelfth International Conference on Learning Representations.
>
> [2] Duan, J., Cheng, H., Wang, S., Zavalny, A., Wang, C., Xu, R., ... & Xu, K. (2024, August). Shifting attention to relevance: Towards the predictive uncertainty quantification of free-form large language models. In Proceedings of the 62nd Annual Meeting of the Association for Computational Linguistics (Volume 1: Long Papers) (pp. 5050-5063).
>
> [3] Lin, S., Hilton, J., & Evans, O. Teaching Models to Express Their Uncertainty in Words. Transactions on Machine Learning Research.
>
> [4] Yona, G., Aharoni, R., & Geva, M. (2024). Can Large Language Models Faithfully Express Their Intrinsic Uncertainty in Words?. arXiv preprint arXiv:2405.16908.
>
> [5] Tian, K., Mitchell, E., Zhou, A., Sharma, A., Rafailov, R., Yao, H., ... & Manning, C. D. Just Ask for Calibration: Strategies for Eliciting Calibrated Confidence Scores from Language Models Fine-Tuned with Human Feedback. In The 2023 Conference on Empirical Methods in Natural Language Processing.
>
> [6] Xiong, M., Hu, Z., Lu, X., LI, Y., Fu, J., He, J., & Hooi, B. Can LLMs Express Their Uncertainty? An Empirical Evaluation of Confidence Elicitation in LLMs. In The Twelfth International Conference on Learning Representations.
>
> [7] Wang, P., Li, L., Chen, L., Cai, Z., Zhu, D., Lin, B., ... & Sui, Z. (2023). Large language models are not fair evaluators. arXiv preprint arXiv:2305.17926.
>
> [8] Koo, R., Lee, M., Raheja, V., Park, J. I., Kim, Z. M., & Kang, D. (2023). Benchmarking cognitive biases in large language models as evaluators. arXiv preprint arXiv:2309.17012.
>
> [9] Liu, Y., Yang, T., Huang, S., Zhang, Z., Huang, H., Wei, F., ... & Zhang, Q. (2024, May). Calibrating LLM-Based Evaluator. In Proceedings of the 2024 Joint International Conference on Computational Linguistics, Language Resources and Evaluation (LREC-COLING 2024) (pp. 2638-2656).

---

> ### Comment · Reviewer_BN93 · 2024-11-26
>
> Thanks for your response, the additional experiments are great and essential, please include them in the paper.
> However, I still believe there is a significant gap between the quality of model evaluation and the output probability, which the author has somewhat confused in the paper.
> Consequently, I will raise the score to 5, and that will be my final score.

---

### Official Review · Reviewer_mJNs · 2024-11-04

**Soundness:** 3
**Presentation:** 3
**Contribution:** 3
**Rating:** 6
**Confidence:** 3

**Summary:**

The paper probes the stability of LLMs-as-Judges by leveraging their internal uncertainties and the confidence levels of the responses of models being evaluated. Single response and pairwise comparisons schemes are compared, where confidence levels are computed based on the token probabilities.  The empirical study includes a variety of LLM evaluators, datasets, and assesses several prompting techniques, including CoT method. The work shows how different prompting strategies can diminish evaluation uncertainty and how uncertainty can be  utilized to improve the reliability evaluators in OOD scenarios. An uncertainty-aware evaluator "ConfiLM" is fine-tuned on a human-annotated dataset, then tested on a new OOD test set sampled from the 2024 Olympics. Findings suggest that incorporating uncertainty significantly improves evaluation accuracy under OOD context.

**Strengths:**

The paper addresses a timely and increasingly relevant issue, the stability of LLMs-as-Judges. While the use of log probabilities as a measure of confidence is not novel in itself, the original contribution lies in its application within the context of evaluators. It investigates methods on how to improve the evaluators confidence and even recognize incorrect responses.
The work is significant for its practical approach to improving LLM evaluator reliability, especially in OOD scenarios. The findings could impact the deployment and trustworthiness of automated evaluators in various applications.

The paper is clearly written, with a well-organized structure that helps with following the multiple methodologies and conducted experiments. Limitations are also acknowledged.

**Weaknesses:**

- Previous studies, such as those by Lyu et al. [1], have shown that log probabilities do not always correlate with human preferences. This raises concerns about the reliability of using these metrics in LLM-as-Judge systems, which have become popular for their scalability and cost-effectiveness in automating evaluations that traditionally relied on human feedback. The paper could further explore how improving model confidence affects agreement with human evaluators. either via CoT or fine-tuning,

- The methodology assumes access to internal model probabilities, which might not be feasible with proprietary models. This limitation, acknowledged in the paper, constrains the generalizability and scalability of the proposed methods.

[1] Lyu, Chenyang, Wu, Minghao  and Aji, Alham. Beyond Probabilities: Unveiling the Misalignment in Evaluating Large Language Models. In Proceedings of the 1st Workshop on Towards Knowledgeable Language Models (KnowLLM 2024)

**Questions:**

1. The paper explains how to compute evaluator and response confidence separately. it was not initially clear that the response confidence is solely utilized for the fine-tuning of ConfiLM and not combined with the evaluator confidence.
2. Could the authors clarify how the results from the human annotators are aggregated?
3. In situations involving pairwise comparisons of models from two distinct families, the log probabilities can span different ranges. Did you consider any recalibration methods to address this issue?
3. Does improving the confidence metrics within LLM evaluators guarantee improvements in their practical evaluation abilities? is this a to be expected outcome?

---

> ### Author Response · Authors · 2024-11-21
> **Response to Reviewer mJNs**
>
> Thank you for your valuable feedback. Please see our responses below.
>
> ### **1. The details of human annotation. (Question 2)**
>
>   - To ensure the quality and consistency of the annotations, we first selected 100 samples from the dataset for preliminary annotation by two of the authors. This process facilitated the development of a well-defined annotation guideline. Then, we hired three PhD-level human annotators who are fluent in English from an annotation company to annotate all samples (both the fine-tuning set and the test set) in two rounds.
>
>   - (1) In the first round, two annotators were asked to label each sample based on the established annotation guidelines. (2) In the second round, a third annotator reviewed samples where disagreements arose and provided a label. The final label for each sample was determined through majority voting. During the annotation process, samples unanimously deemed low quality or difficult to evaluate by the annotators were excluded. We have added the details to Section B for clarity.
>
> ### **2. What is the relation between evaluation uncertainty and evaluation quality? Does improving confidence affect evaluation performance? (Weakness 1 and  Question 4)**
>
>   - To address this question, we analyze the average accuracy of judgments made by six LLM-based evaluators across different confidence intervals. The experiments were conducted on our human-annotated test set (Olympic 2024). The results are presented in Table 1. This table demonstrates **a positive correlation between evaluation confidence and evaluation accuracy**. Specifically, when evaluation confidence is low, the accuracy of judgments across evaluators is generally lower. As evaluation confidence increases, judgment accuracy improves steadily, reaching peak performance in high-confidence intervals (e.g., [0.8, 1.0)). This indicates that models are more reliable in performing evaluation tasks when outputting with higher confidence.
>
>   - Table 1: The relation between evaluation confidence and evaluation accuracy on Olympic 2024.
>
>       |       Evaluator      | [0.0, 0.2) | [0.2, 0.4) | [0.4, 0.6) | [0.6, 0.8) | [0.8, 1.0) |
>       |:--------------------:|:----------:|:----------:|:----------:|:----------:|:----------:|
>       |        GPT-4o        |   0.000    |   0.250    |   0.333    |    0.625   |   0.684    |
>       |      GPT-4o-mini     |   0.000    |   0.333    |   0.222    |    0.625   |   0.721    |
>       |     GPT-3.5-Turbo    |   0.125    |   0.333    |   0.400    |    0.556   |   0.634    |
>       | Llama-3-70B-Instruct |   0.000    |   0.000    |   0.200    |    0.364   |   0.680    |
>       | Llama-2-70B-Instruct |   0.000    |   0.000    |   0.000    |    0.267   |   0.579    |
>       |  Qwen2-72B-Instruct  |   0.000    |   0.500    |   0.571    |    0.600   |   0.668    |

---

> ### Author Response · Authors · 2024-11-21
>
> ### **3. Can we obtain consistent experimental findings if different methods of measuring uncertainty are used? (Weakness 2)**
>   - In the original submission, we used generation logits as a proxy for model confidence. To investigate whether different definitions of uncertainty impact the final findings, we conducted additional experiments under a pairwise comparison setting on the MT-Bench. These experiments involved two commonly used confidence estimation methods: (1) **Verbalization-based confidence**, where we prompted LLMs to directly output calibrated confidence scores along with their responses [1][2]; (2) **Consistency-based confidence**, which involved generating 5/10/20 responses to the same question and measuring their consistency as a proxy for confidence [3][4]. For these experiments, we set the sampling temperature to 0.7.
>   - In the experiments, the evaluation subjects were Llama2-7B-Instruct and Llama2-13B-Instruct. The confidence estimation results are presented in Table 2. **Based on the analysis of these results, we observed that the evaluation confidence obtained using different confidence estimation methods follows the same patterns.** This further supports the conclusions drawn in the original submission: (1) LLM evaluators exhibit varying levels of uncertainty. (2) Evaluations within the same model family demonstrate higher evaluation confidence.
>   - Table 2: The evaluation confidence estimation results with different estimation methods.
>
>       |       Evaluator      | Logit-based  | Verbalization-based  | Consistency-5 | Consistency-10 | Consistency-20 |
>       |:--------------------:|:------------:|:--------------------:|:-------------:|:--------------:|:--------------:|
>       |        GPT-4o        |     0.699    |         0.764        |     0.751     |      0.725     |      0.696     |
>       |      GPT-4o-mini     |     0.776    |         0.725        |     0.771     |      0.736     |      0.735     |
>       |     GPT-3.5-Turbo    |     0.848    |         0.755        |     0.828     |      0.790     |      0.804     |
>       | Llama-3-70B-Instruct |     0.791    |         0.808        |     0.846     |      0.833     |      0.778     |
>       | Llama-2-70B-Instruct |     0.908    |         0.856        |     0.911     |      0.915     |      0.891     |
>       |  Qwen2-72B-Instruct  |     0.762    |         0.730        |     0.765     |      0.717     |      0.657     |
>
> ### **4. In situations involving pairwise comparisons of models from two distinct families, do we need to align the response uncertainties? (Question 3)**
>
>   - Thank you for pointing this out. We agree with your view that since candidate models from different families naturally have different distributions of response confidence, it is important to consider this difference during pairwise comparisons. **To mitigate this issue, we have taken the following measures:**
>     - (1) In experiments investigating the properties of uncertainty (Section 4), we selected candidate models belonging to the same family: Llama-7B and Llama-13B, to eliminate the influence of this issue on the main conclusions of the paper.
>     - (2) When annotating fine-tuning and test data from different model families (Section 5.1), we asked human annotators to focus on the difference between the two response confidences rather than their specific values.
>     - (3) When training/testing ConfiLM using response confidences from different model families, we verbalize each instance's u₁ and u₂ into natural language statements. Response confidences within the same numerical interval are mapped to the same statement—for example, [0.6, 0.7) → "Slightly confident", which also encourages the evaluator to focus on differences between the two response confidences.
>
> ### **5. Why not use fine-tuning to mitigate response uncertainty? (Question 1)**
>   - Response uncertainty does not necessarily need to be mitigated through fine-tuning. As demonstrated in Section 4 of our paper, using specialized output formats, such as Chain-of-Thought (CoT) reasoning, can effectively reduce evaluation uncertainty without requiring fine-tuning. During the fine-tuning and testing of ConfiLM, we added the response confidences of candidate models into the prompt to explicitly inform the evaluator about the response uncertainty. We will stress this in Section 5 for clarity.
>
> **We appreciate the practical tips you provided, which have helped us improve our paper. We look forward to further communication with you.**

---

> ### Author Response · Authors · 2024-11-21
>
> [1] Lin, S., Hilton, J., & Evans, O. Teaching Models to Express Their Uncertainty in Words. Transactions on Machine Learning Research.
>
> [2] Yona, G., Aharoni, R., & Geva, M. (2024). Can Large Language Models Faithfully Express Their Intrinsic Uncertainty in Words? arXiv preprint arXiv:2405.16908.
>
> [3] Tian, K., Mitchell, E., Zhou, A., Sharma, A., Rafailov, R., Yao, H., ... & Manning, C. D. Just Ask for Calibration: Strategies for Eliciting Calibrated Confidence Scores from Language Models Fine-Tuned with Human Feedback. In The 2023 Conference on Empirical Methods in Natural Language Processing.
>
> [4] Xiong, M., Hu, Z., Lu, X., LI, Y., Fu, J., He, J., & Hooi, B. Can LLMs Express Their Uncertainty? An Empirical Evaluation of Confidence Elicitation in LLMs. In The Twelfth International Conference on Learning Representations.

---

### Official Review · Reviewer_Yumv · 2024-11-04

**Soundness:** 3
**Presentation:** 3
**Contribution:** 3
**Rating:** 6
**Confidence:** 4

**Summary:**

While the importance of measuring and handling uncertainty and confidence with LLMs is a known and well-researched topic, this paper seems to be the first to analyse uncertainty and evaluation confidence within the particular 'LLM-as-a-judge' (LLMJ) setting. It runs a series of experiments that yield a series of empirical observations related to uncertainty of LLMs when used as evaluators, and then it aims to show that incorporating uncertainty modeling directly into the process of creation of LLM-based evaluators can yield more confident and better-performing evaluators. Evaluations are conduced with a reasonable number of well-known open-weights and API-gated models, within (i) direct-scoring and (ii) pairwise preference setups. The authors then propose an uncertainty-aware fine-tuning process and create such an uncertainty-aware LLMJ model ConfiLM which shows benefits in evaluations on OOD data.

**Strengths:**

- The first paper that empirically examines the role of uncertainty within the context of LLMJ, offering a series of empirical findings that motivate the creation of an uncertainty-aware LLM evaluator.
- Despite the large number of experiments, it is quite clear what the focus is in each of the experiments, and it's good to see some intuitions properly empirically verified.
- The choice of underlying model is quite good, which helps with generalising some of the empirical findings.

**Weaknesses:**

- The paper adopts a simplified view on recent 'LLM-as-a-Judge' (LLMJ) literature, skipping some related work that aimed to improve confidence of LLMJ models via calibration (https://openreview.net/pdf?id=L3FHMoKZcS) or better prompt optimization (PO; https://arxiv.org/pdf/2406.11370). Improving LLMJ methods requires a multi-component/multi-aspectual approach, which requires combinations of prompt optimisation, calibration and uncertainty mitigation strategies, and the paper would be much stronger with additional experiments that aim to approach the problem in full scope, without reducing it only to uncertainty mitigation. NOTE: This has been corrected in the author response
- This also means that it would be extremely useful to measure how uncertainty/confidence levels change when different bias mitigation or calibration or PO methods get applied. Do we need to handle uncertainty to the same extend and with the same benefits when it gets combined with other methods?
- Speaking of simplifying the experimental protocol too much, there are also many potential approaches to measuring uncertainty, and taking only the logit as the proxy towards uncertainty, while grounded in some related literature, might still be too simple. I would recommend exploring different ways of measuring uncertainty (e.g., see https://arxiv.org/pdf/2404.15993 or https://arxiv.org/abs/2307.10236) and checking whether different definitions of uncertainty also have impact on the final findings (and to what extent). The decision on how we measure/capture uncertainty is a big assumption that seems to stay under the radar of this work. NOTE: This has also been mitigated with the author response.
- Evaluation should get increased to more and more standard LLMJ datasets, including e.g., RewardBench and standard summarization and chat datasets used in prior work (SummEval, HANNA, etc).
- Minor: the paper spends too much introducing the basics of evaluation with LLMs (e.g., a large part of RW is too long) - that paper space could have been better used for some finer-grained experiments (e.g., understanding the relationships between calibration and certainty or introducing another way to measure uncertainty or some other future work ideas now listed in lines 521-529).

**Questions:**

- Is there any relationship and/or correlation between uncertainty in the responses of the evalauted mopdels and uncertainty of the evaluator models? How does uncertainty of the evaluated model reflect on the final scores?
- I liked the finding showing how evaluators are more confident with responses coming from models within the same family. Is there a way to also mitigate this 'uncertainty bias' through fine-tuning as well?
- It seems that 'Ties' cause a lot of trouble with confidence - how would discarding the 'Tie' option from the preferences output impact the results and the main findings? I would recommend adding an experiment checking on this.
- ConfiLM shows benefits with OOD data. Are there any benefits with in-domain or related-domain data or is the approach useful only in OOD scenarios?

---

> ### Author Response · Authors · 2024-11-21
> **Response to Reviewer Yumv**
>
> Thank you for your thoughtful and valuable comments. Below are our responses to your specific points and suggestions:
>
> ### **1. Explore different experimental settings to verify the validity of our findings (Weaknesses 3, 4).**
>
>   - Different ways of measuring uncertainty:
>     - In the original submission, we used generation logits as a proxy for model confidence. To investigate whether different definitions of uncertainty impact the final findings, we conducted additional experiments under a pairwise comparison setting on the MT-Bench. These experiments involved two commonly used confidence estimation methods: (1) **Verbalization-based confidence**, where we prompted LLMs to directly output calibrated confidence scores along with their responses [1][2]; (2) **Consistency-based confidence**, which involved generating 5/10/20 responses to the same question and measuring their consistency as a proxy for confidence [3][4]. For these experiments, we set the sampling temperature to 0.7.
>     - In the experiments, the evaluation subjects were Llama2-7B-Instruct and Llama2-13B-Instruct. The confidence estimation results are presented in Table 1. **Based on the analysis of these results, we observed that the evaluation confidence obtained using different confidence estimation methods follows the same patterns.** This further supports the conclusions drawn in the original submission: (1) LLM evaluators exhibit varying levels of uncertainty. (2) Evaluations within the same model family demonstrate higher evaluation confidence.
>     - Table 1: The evaluation confidence estimation results with different estimation methods.
>
>         |       Evaluator      | Logit-based  | Verbalization-based  | Consistency-5 | Consistency-10 | Consistency-20 |
>         |:--------------------:|:------------:|:--------------------:|:-------------:|:--------------:|:--------------:|
>         |        GPT-4o        |     0.699    |         0.764        |     0.751     |      0.725     |      0.696     |
>         |      GPT-4o-mini     |     0.776    |         0.725        |     0.771     |      0.736     |      0.735     |
>         |     GPT-3.5-Turbo    |     0.848    |         0.755        |     0.828     |      0.790     |      0.804     |
>         | Llama-3-70B-Instruct |     0.791    |         0.808        |     0.846     |      0.833     |      0.778     |
>         | Llama-2-70B-Instruct |     0.908    |         0.856        |     0.911     |      0.915     |      0.891     |
>         |  Qwen2-72B-Instruct  |     0.762    |         0.730        |     0.765     |      0.717     |      0.657     |
>
>   - Discarding the 'Tie' option from the preferences output.
>     - To address this question, we conducted additional experiments on the MT-Bench by removing the 'Tie' option from the preferences output. The results are presented in Table 2. We observed that discarding the 'Tie' option does help evaluators reduce some evaluation uncertainty. However, a considerable degree of evaluation uncertainty remains, and the observed patterns align with the conclusions drawn in the original submission.
>     - Table 2: The evaluation confidence of 6 LLM-based evaluators.
>         |       Evaluator      | Result from the original submission | Results with the 'Tie' option discarded  |
>         |:--------------------:|:-----------------------------------:|:----------------------------------------:|
>         |        GPT-4o        |                0.699                |                   0.824                  |
>         |      GPT-4o-mini     |                0.776                |                   0.806                  |
>         |     GPT-3.5-Turbo    |                0.848                |                   0.816                  |
>         | Llama-3-70B-Instruct |                0.791                |                   0.823                  |
>         | Llama-2-70B-Instruct |                0.908                |                   0.920                  |
>         |  Qwen2-72B-Instruct  |                0.762                |                   0.850                  |
>
> These two experiments, conducted under different experimental conditions, further demonstrate the validity of the conclusions presented in this study. The results will be integrated into Section 3. **We sincerely thank the reviewers for suggesting these experimental setups, which have helped us enhance the overall quality of the paper.**

---

> ### Author Response · Authors · 2024-11-21
>
> ### **2. Exploratory experiments conducted based on the findings of this study (Questions 1, 3, 4).**
>   - What is the relation between evaluation confidence and evaluation accuracy?
>     - To address this question, we analyzed the average accuracy of judgments made by six LLM-based evaluators across different confidence intervals. The results are presented in Table 3. **This table demonstrates a positive correlation between evaluation confidence and accuracy.** Specifically, when evaluation confidence is low, the accuracy of judgments across evaluators is generally lower. As evaluation confidence increases, judgment accuracy improves steadily, reaching peak performance in high-confidence intervals (e.g., [0.8, 1.0)). This indicates that models are more reliable in performing evaluation tasks when evaluating with higher confidence.
>     - Table 3: The relation between evaluation confidence and evaluation accuracy on Olympic 2024.
>
>         |       Evaluator      | [0.0, 0.2) | [0.2, 0.4) | [0.4, 0.6) | [0.6, 0.8) | [0.8, 1.0) |
>         |:--------------------:|:----------:|:----------:|:----------:|:----------:|:----------:|
>         |        GPT-4o        |   0.000    |   0.250    |   0.333    |    0.625   |   0.684    |
>         |      GPT-4o-mini     |   0.000    |   0.333    |   0.222    |    0.625   |   0.721    |
>         |     GPT-3.5-Turbo    |   0.125    |   0.333    |   0.400    |    0.556   |   0.634    |
>         | Llama-3-70B-Instruct |   0.000    |   0.000    |   0.200    |    0.364   |   0.680    |
>         | Llama-2-70B-Instruct |   0.000    |   0.000    |   0.000    |    0.267   |   0.579    |
>         |  Qwen2-72B-Instruct  |   0.000    |   0.500    |   0.571    |    0.600   |   0.668    |
>
>   - ConfiLM shows benefits with OOD data. How does it perform on in-domain(ID) data?
>     - In the original submission, we fine-tuned an uncertainty-aware LLM evaluator named ConfiLM, which leverages the response confidence of candidate models to enhance evaluation capability for OOD data. To investigate ConfiLM's performance on ID data, we re-split its fine-tuning dataset, selecting 94 human-annotated instances as an in-domain test set, **Alpaca-94**. Based on the remaining 600 fine-tuning instances, we re-trained the models using the same experimental setup as in the original submission, producing **ConfiLM-600** and **Llama-3-8B-Instruct-Finetune-600**. Their evaluation performance on Alpaca-94 (ID) and Olympic 2024 (OOD) is reported in Table 4.
>     - Experimental results demonstrate that incorporating uncertainty as auxiliary information significantly enhances the performance of LLM evaluators in OOD scenarios. While ConfiLM-600's advantage is reduced in ID scenarios, it still achieves evaluation performance comparable to Llama-3-8B-instruct-finetune-600.
>
>     - Table 4: Evaluation of Alpaca-94 and Olympic 2024.
>         |              | ConfiLM-600 | Llama-3-8B-instruct-finetune-600 | Llama-3-8B-instruct |
>         |:------------:|:-----------:|:--------------------------------:|:-------------------:|
>         |   Alpaca-94  |    0.581    |               0.585              |        0.518        |
>         | Olympic 2024 |    0.577    |               0.535              |        0.519        |
>
>   - Is there a correlation between response confidence and evaluation confidence?
>     - To answer this question, we calculated the Pearson correlation coefficient under the single-answer grading setting on the MT-Bench. The results are presented in Table 5. From the results, we observed that when evaluators were not explicitly informed about response confidence, there was no consistent correlation between response confidence and evaluation confidence across the six evaluators. However, after incorporating response confidence into the prompt, a weak positive correlation emerged between response confidence and evaluation confidence for all six evaluators.
>     - Table 5: Pearson correlation between response confidence and evaluation confidence.
>
>         |       Evaluator      | Default prompt | Adding response confidence to the prompt |
>         |:--------------------:|:--------------:|:----------------------------------------:|
>         |        GPT-4o        |     -0.039     |                   0.346                  |
>         |      GPT-4o-mini     |     -0.167     |                   0.140                  |
>         |     GPT-3.5-Turbo    |     -0.027     |                   0.241                  |
>         | Llama-3-70B-Instruct |      0.135     |                   0.345                  |
>         | Llama-2-70B-Instruct |      0.244     |                   0.412                  |
>         |  Qwen2-72B-Instruct  |      0.034     |                   0.173                  |

---

> ### Author Response · Authors · 2024-11-21
>
> ### **3. Is there a way to mitigate self-preference bias through fine-tuning? (Question 2)**
>   - We appreciate your constructive suggestions. The conclusion of this study highlights the potential risks of self-preference bias when evaluators from the same model family are used, which could lead to biased evaluations. We recommend addressing this issue during the fine-tuning process by applying the following methods: (1) Introduce a regularization term to penalize evaluations with disproportionately high confidence for similar models. (2) Use adversarial examples, where related models generate suboptimal outputs but mimic the evaluator's style, to challenge and improve the evaluator's objectivity.
>
> ### **4. Presentation (Weaknesses 1, 2, 5)**
>   - We sincerely thank you for your suggestions regarding future directions for this work. We agree that improving LLMJ methods requires a comprehensive consideration of combinations of prompt optimization, calibration, and uncertainty mitigation strategies. Following your advice, **we have revised the discussion section to include relevant works**, such as **Batch Calibration** (https://openreview.net/pdf?id=L3FHMoKZcS), **ClaPS** (https://arxiv.org/abs/2310.12774), and **ZEPO** (https://arxiv.org/pdf/2406.11370).
>   - We will follow your valuable advice to revise the structure of the paper and submit a new version soon.
>
> [1] Lin, S., Hilton, J., & Evans, O. Teaching Models to Express Their Uncertainty in Words. Transactions on Machine Learning Research.
>
> [2] Yona, G., Aharoni, R., & Geva, M. (2024). Can Large Language Models Faithfully Express Their Intrinsic Uncertainty in Words?. arXiv preprint arXiv:2405.16908.
>
> [3] Tian, K., Mitchell, E., Zhou, A., Sharma, A., Rafailov, R., Yao, H., ... & Manning, C. D. Just Ask for Calibration: Strategies for Eliciting Calibrated Confidence Scores from Language Models Fine-Tuned with Human Feedback. In The 2023 Conference on Empirical Methods in Natural Language Processing.
>
> [4] Xiong, M., Hu, Z., Lu, X., LI, Y., Fu, J., He, J., & Hooi, B. Can LLMs Express Their Uncertainty? An Empirical Evaluation of Confidence Elicitation in LLMs. In The Twelfth International Conference on Learning Representations.

---

> ### Comment · Reviewer_Yumv · 2024-11-25
> **Many thanks for the response!**
>
> The new results and additional discussion points increased the paper potential, which is reflected in my revised score. I still have doubts on the impact of the paper in light of very similar related work (which is also the main reason why I wouldn't go beyond 6 as my final recommendation score).
>
> Also, I think that besides only discussing related work suggested in my review, the authors should also ideally compare with these approaches (e.g., applying batch calibration and/or prompt optimization) and how these approaches affect uncertainty (e.g., 'before' vs 'after' comparisons).

---

### Author Response · Authors · 2024-12-04
**General Responses and Summary of Revisions**

We sincerely thank all reviewers for their careful and constructive feedback, which has helped us significantly improve our work. We are encouraged by the reviewers' recognition of several key strengths:

- "_The first paper that empirically examines the role of uncertainty within the context of LLMJ, ... it's good to see some intuitions properly empirically verified._" (Reviewer Yumv)
- "_An in-depth evaluation uncertainty of various models ... I appreciate the dataset and model created in this paper. These can surely benefit the community in future research in this domain._" (Reviewer 2uFv)
- "_The original contribution lies in its application within the context of evaluators ... The work is significant for its practical approach to improving LLM evaluator reliability, especially in OOD scenarios._" (Reviewer mJNs)
- "_This paper conducts extensive empirical analyses across multiple models and datasets, offering results that can help researchers deepen their understanding of various models as evaluators._" (Reviewer BN93)

Based on their constructive comments, we have made substantial improvements to strengthen our paper's contributions and address the limitations. The major enhancements include:
- **Additional Experimental Results.** We have expanded our experimental scope to validate our findings across different settings, offering  an analysis of uncertainty under different quantification methods (Section 3, Appendix B.1) and exploring the relation between evaluation confidence and accuracy (Appendix B.2). Furthermore, we provided experimental results about the effectiveness of verbalization (Table 21) and the in-domain evaluation performance of ConfiLM (Appendix B.3). These additional results further validate the conclusions drawn in our paper.
- **More Details about the Proposed Dataset and Model.** We have provided more specific details regarding the distribution of response confidence (Figure 10), comparative performance results across different fine-tuning hyperparameters (Figure 11), and the process of human annotation (Appendix C).
- **Extended Discussion.** We have enriched our paper with a more comprehensive discussion on related studies (Section 2), an explanation for score fluctuations (Section 4.5), and considerations for future research directions (Section 6).

We thank the reviewers again for their valuable input that has helped us improve the paper. We believe these additions have fully realized the potential of our research contribution to the field of model-based LLM evaluations.

---

### Meta-Review · Area_Chair_B6A4 · 2024-12-19

**Metareview:**

**Summary**

This paper falls into the category of evaluating LLMs with LLMs. It enhance the area by proposing a model including uncertainity.


**Strengths**

- The paper aggregates several tiny established yet undocumented/poorly documented ideas into one place as a formal work (reviewer 2uFv)

**Weaknesses**

- The idea of LLM-as-a-Judge is somehow simplified as discussed by the reviewer Yumv.
- Uncertainty is not fully explored, and it is over simplified (reviewer Yumv).
- The method seems to be applicable only for non-proprietary models (Reviewer mJNs) as relies on internal probabilities. This is a major limit.

**Additional Comments On Reviewer Discussion:**

Reviewers have partially interacted and (possibly) modified their scores.

---

### Decision · Program_Chairs · 2025-01-22

Accept (Poster)